# The role of platelets in mediating a response to human influenza infection

Milka Koupenova [1], Heather A. Corkrey[1], Olga Vitseva[1], Giorgia Manni[2], Catherine J. Pang [3], Lauren Clancy [1], Chen Yao [4], Jeffrey Rade[1], Daniel Levy[4], Jennifer P. Wang [3], Robert W. Finberg[3], Evelyn A. Kurt-Jones[3] & Jane E. Freedman[1]

Influenza infection increases the incidence of myocardial infarction but the reason is unknown. Platelets mediate vascular occlusion through thrombotic functions but are also recognized to have immunomodulatory activity. To determine if platelet processes are activated during influenza infection, we collected blood from 18 patients with acute influenza infection. Microscopy reveals activated platelets, many containing viral particles and extracellular-DNA associated with platelets. To understand the mechanism, we isolate human platelets and treat them with influenza A virus. Viral-engulfment leads to C3 release from platelets as a function of TLR7 and C3 leads to neutrophil-DNA release and aggregation. TLR7 specificity is confirmed in murine models lacking the receptor, and platelet depletion models support platelet-mediated C3 and neutrophil-DNA release post-influenza infection. These findings demonstrate that the initial intrinsic defense against influenza is mediated by platelet–neutrophil cross-communication that tightly regulates host immune and complement responses but can also lead to thrombotic vascular occlusion.

[1] Department of Medicine, Division of Cardiovascular Medicine, 368 Plantation Street, Worcester, MA 01605, USA. [2] Department of Internal Medicine, Section of Internal and Cardiovascular Medicine, University of Perugia, Polo Unico Sant'Andrea delle Fratte, Perugia 06132, Italy. [3] Department of Infectious Disease and Immunology, University of Massachusetts Medical School, 55 Lake Street, Worcester, MA 01605, USA. [4] Population Sciences Branch, National Heart, Lung, and Blood Institute, National Institutes of Health, 31 Center Dr, Bethesda, MD 20824, USA. Correspondence and requests for materials should be addressed to M.K. (email: milka.koupenova@umassmed.edu)

Heart disease is the leading cause of morbidity and mortality in the US with 735,000 people per year experiencing myocardial infarction (MI)[1]. Viral infections such as influenza increase the incidence of acute MI within the first 7 days after detection of influenza A or B, whereas no increased incidence is observed after day 7[2]. Meta-analysis of case-control studies finds that vaccination for influenza is comparable to current therapies for secondary prevention of acute MI such as statins or antihypertensive medications[3]. The exact mechanism by which influenza contributes to acute coronary syndromes and cardiovascular disease is not understood.

In humans, platelets are central to the process of thrombosis and uncontrolled platelet activation is the main factor in unstable coronary syndromes and acute MI[1,4]. In addition to their role in thrombosis, platelets significantly contribute to the immune response in various types of infections[5]. During the initial stages of infection, platelets engage and form heterotypic aggregates with neutrophils[6,7]. Heterotypic aggregates between platelets and neutrophils are observed during Gram-positive bacterial infections, with Gram-negative bacterial components as well as during infections with single-stranded viruses such as encephalomyocarditis virus[6,8,9]. Platelets exhibit a critical adaptive immune function by forming platelet–bacterial complexes that slow bacterial clearance and increase antibacterial immunity[10].

Influenza is a single-stranded RNA (ssRNA) virus that is recognized by cell-surface sialic acid which serves as an influenza receptor. Influenza causes productive infection in lung epithelial cells which can lead to various degrees of severity of illness. In humans, one pattern recognition receptor that mediates the initial response to ssRNA viral nucleic acids is Toll-like receptor 7 (TLR7). Once activated, TLR7 elicits a cascade of signaling events that lead to primary interferon secretion and activation of the immune system. Platelets express TLR7, although not all platelets in an individual express TLR7 at any given time[6,11]. Activation of TLR7 in platelets leads to surface expression of alpha granule proteins, P-selectin, and CD40L and a consequent increase in interaction with neutrophils without leading to a direct platelet-mediated prothrombotic effect[6]. It is unclear if downstream activation of the immune system mediated by platelet-TLR7 leads to platelet-dependent thrombosis, which could potentially increase the risk for MI.

The neutrophil is the major leukocyte that mediates the initial response to pathogens. Neutrophils are the most prevalent leukocyte in humans and, in the presence of influenza, isolated neutrophils exhibit decreased viability, increased respiratory burst, and accelerated apoptosis[12,13]. Additionally, influenza-stimulated neutrophils can also release their DNA in a process termed NETosis[13]. This DNA release is thought to benefit the host and provide protection during viral challenge but can also be highly prothrombotic. Consistently, neutrophils from the lesion site involved in the initial acute MI are highly activated, form platelet–neutrophil aggregates and can lead to NET burden that is a predictor of ST-segment resolution and extent of MI[14]. Platelets are known to reduce the time to NETosis through engagement of TLR2 and TLR4[15,16]. It is unknown whether platelets contribute to overall neutrophil activation through other TLRs (e.g., TLR7 which becomes activated by viral ligand) during influenza infection or if the platelet–neutrophil relationship becomes pathologically imbalanced during infection.

In addition to the TLR-mediated response during infection, innate immunity includes activation of the complement system. The complement system incorporates three distinct pathways leading to opsonization of pathogens, chemotaxis of inflammatory cells and lysis of infected cells, ultimately leading to the removal of immune complexes, apoptotic cells and cell debris. In mice, complement component C3 is required for protection against influenza and for proper viral clearance[17]. The mechanism by which the complement and the TLR systems cross-communicate, as well as the impact of complement-platelet-TLR interactions during influenza infection in humans, is not known.

In this study, we sought to evaluate the effect of TLR7 and the complement system on platelets during influenza infection and their possible impact on augmenting thrombosis. Our data demonstrate that platelet-TLR7-driven responses lead to C3 release during influenza infection and consequently, C3 augments the release of DNA from neutrophils and promotes the formation of platelet–neutrophil aggregates which may contribute to influenza-mediated increased risk of MI.

## Results

**Platelet morphology and DNA release in influenza patients.** To study the impact of influenza infection on platelets we asked if platelets change morphologically in the human circulation in the setting of influenza. We collected blood from 18 patients with acute influenza A H1N1, influenza A H3N2, or influenza B over a period of 3 years. Patient characteristics are described in Supplementary Table 1. Confocal microscopy analysis of human blood from infected individuals revealed the presence of platelets with various morphologies, ranging from small platelets without pseudopodia, to spread-out, flattened satellite-like platelets that in some cases have a diameter bigger than 10 µm (Fig. 1a, b, Supplementary Fig. 1).

Surprisingly, the blood of influenza-infected patients contained aggregates of released DNA associated with platelets (Fig. 1c, d, Supplementary Fig. 2a). In certain patients, this DNA co-localized with the neutrophil markers CD66b and myeloperoxidase (MPO) (Supplementary Fig. 2b, c), suggesting the possibility that the released DNA is due to NETosis. Plasma of influenza-infected patients showed elevated levels of elastase and MPO (Supplementary Fig. 2b, c) but did not test positive for nucleosomes (Fig. 1g) or DNA assessed by PicoGreen® (Supplementary Fig. 3). In the influenza-infected patients, certain DNA-releasing neutrophils did not express CD66b (Supplementary Fig. 4), indicating that some of the released DNA cannot be absolutely identified as coming from neutrophils. Lack of CD66b may be due to secretion of this adhesion molecule during infection as previously observed[14,18]. A possible reason for the discrepancy between released DNA observed by microscopy and lack of DNA detection in the plasma is that the platelet–neutrophil aggregates are large and spin down with cell fractions in blood. Alternatively, kinetic differences may exist between protein and DNA clearance in plasma after infection. Also, phorbol myristate acetate (PMA)-NETosis and viral NETosis differ[19] and may not be easily evaluated in plasma, particularly after a double-spin method with citrated blood.

To further evaluate which cell contributes to the DNA release and to quantify the release at the onset of flu infection, blood from healthy donors was incubated with sucrose-purified influenza (WSN/33). The released DNA co-stained with neutrophil marker CD66b and formed aggregates with platelets (Fig. 1h and Supplementary Fig. 5). Of note, the released DNA was measured in blood from healthy donors treated with WSN/33 at constant rotation. Influenza led to a 40% increase of netting neutrophils when compared to control (Fig. 1i). The finding of unattached netting formations was unexpected since free circulating neutrophil–DNA–platelet formations have not been previously described. During intravascular NETosis, neutrophils have been thought to be attached to organ blood vessels[20] and free netting structures have not been detected directly in blood.

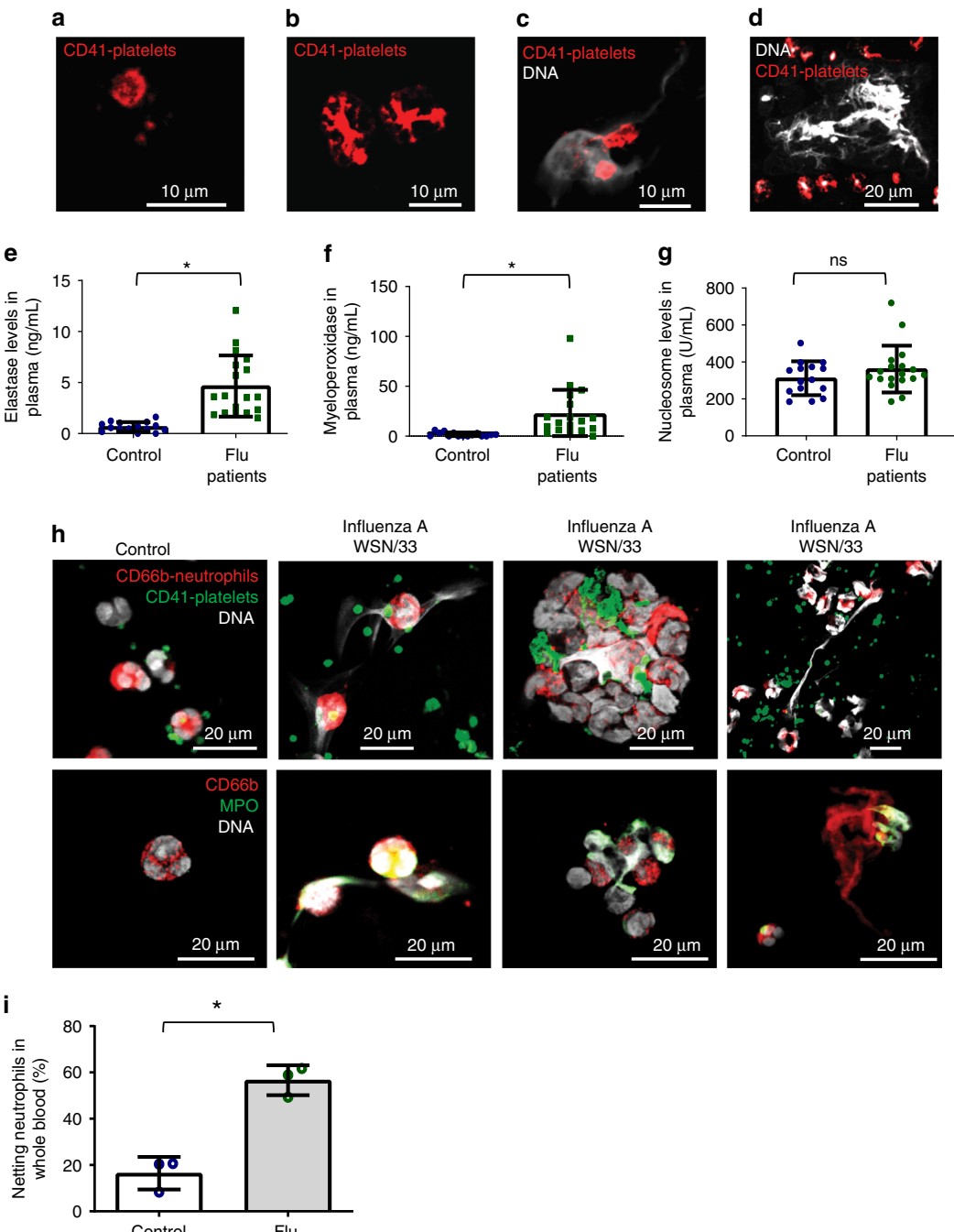

**Fig. 1** Characterization of human blood from influenza-infected patients. **a–d** Blood from influenza-infected patients was fixed after intravenous collection and stained as described in Methods. In all cases DNA was assessed by DAPI. **a** Many platelets appeared similar to those observed in control blood from healthy donors with a size of 2–5 μm. **b** Some platelets from influenza-infected patients had undergone spreading with a distinctive distribution of CD41. **c** Platelets and platelet microparticles associated with DNA (arrow). **d** Spread platelets were found surrounding released DNA with distinct DNA content in their center. Of note, blood from influenza-infected patients and controls in (**a–d**) was not permeabilized. Since formaldehyde can cause certain levels of permeabilization as a function of cross-linking positive staining in control samples for H4 and MPO do not necessarily indicate activation. **e–g** Levels of proteins related to DNA release in the plasma of influenza-infected patients assessed by ELISA. Source data are provided as a Source Data file. **e** neutrophil elastase, **f** myeloperoxidase (MPO), and **g** histone nucleosome core. The graphs represent the average ± SD of healthy donors ($n = 15$) and influenza-infected patients ($n = 18$); significance for (**e–g**) was assessed by Mann–Whitney $U$ test, star symbol (*) indicates $p < 0.0001$. **h, i** To synchronize time of influenza presence as a function of infection and quantify the released DNA, we treated blood from human donors for 30 min with sucrose-purified infectious influenza (WSN/33) at constant rotation and 37 °C. Influenza was used at 1 pfu to 100 platelets. **h** Representative images of blood from 3 donors with influenza and one (out of 3) healthy control and **i** their quantitation. Of note, in certain cases the DNA is not entirely covered with platelets, suggesting differences in kinetics of interaction and/or a physiological relationship that needs further in vivo characterization. Data in graph is represented as average ± SD of $n = 3$ different donors; significance was assessed by unpaired $t$-test (two-tailed value) and star symbol (*) indicates $p = 0.0019$, df = 4

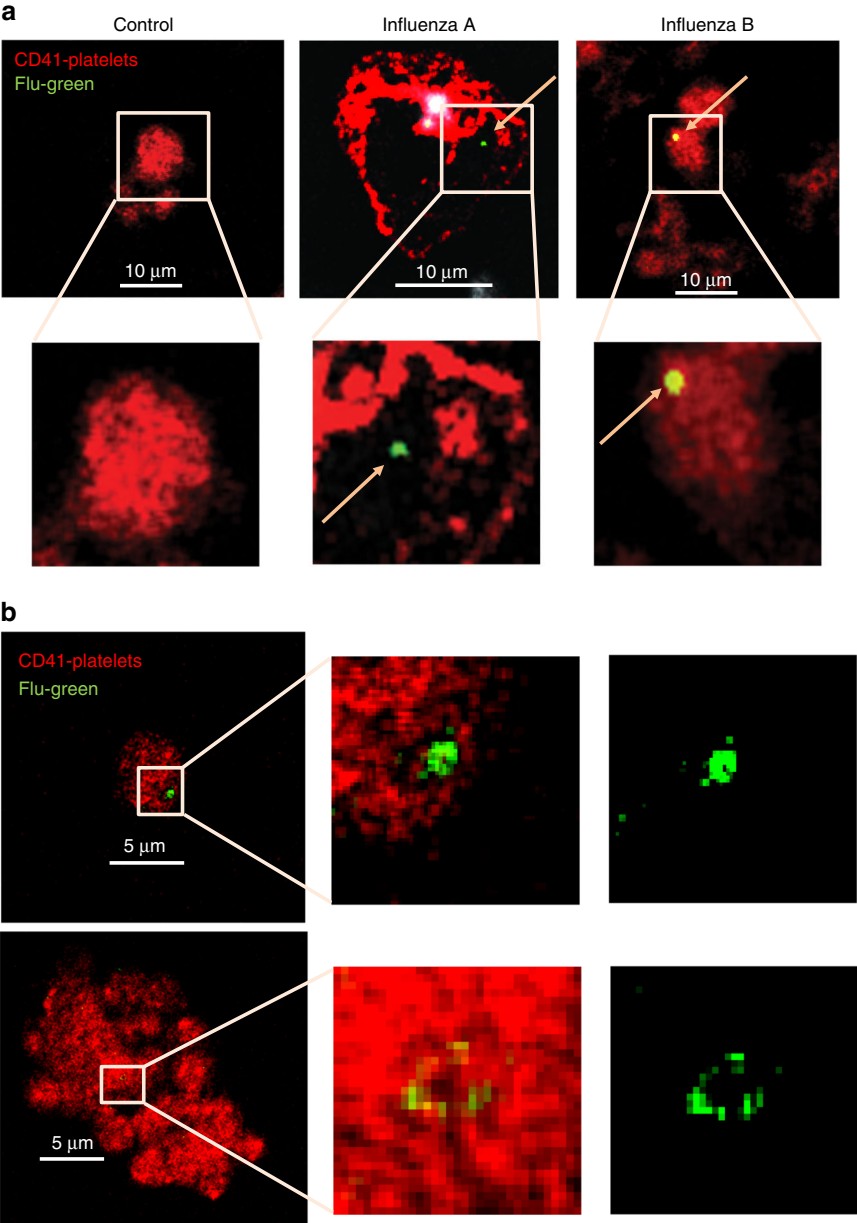

**Fig. 2** Influenza particles are found in platelets by confocal microscopy. 50 µL of intravenous blood drawn from **a** influenza positive patients or healthy donors. Blood was fixed (RBCs were lysed) and samples were later stained with antibodies for the nuclear protein for influenza A (green) or influenza B (green, yellow = green and red combined), for platelets stained with CD41-APC (red), and DNA from neutrophils (white). Arrows point toward the influenza (green) staining. Representative images are shown from $n = 10$ different patients. Confocal microscopy of **b** platelets isolated from healthy donors and incubated with WSN/33, at 1 pfu to 100 platelets, for 30 min, at 1000 rpm, 37 °C. Platelets were fixed, permeabilized, and stained with the same antibodies as in (**a**). Representative images of $n = 4$ (2F, 2M) are shown

**Platelets from infected patients contain influenza particles**. In addition to examining morphological changes, we also evaluated if influenza virions can be detected in association with or within platelets in patients with active influenza infection. For that purpose, we used quantitative PCR (qPCR). Influenza viral RNA detection was variable. Influenza RNA was detected by qPCR in 4 of 18 patients; 10 randomly selected subjects of 18 stained positive for influenza nucleoprotein (NP) in the blood as shown in Fig. 2a (see Supplementary Table 2). In some cases, virions were also found associated with released DNA and platelet CD41 (Supplementary Fig. 6).

To evaluate the time course of influenza internalization into platelets, we isolated platelets from healthy human donors and incubated them with infectious influenza A (WSN/33) virions as a function of time. Following incubation, platelets were fixed and imaged by fluorescent or transmission electron microscopy. We observed influenza attached to platelets as early as 1 min post-incubation. Internalization began 5 min post-incubation, continued at 15 min, and peaked at 30 min (Figs. 2b and 3a, b, Supplementary Fig. 7). Digestion of the virus was observed as early as 15 min (Fig. 3a). Transmission electron microscopy of blood from the patients that tested positive by qPCR showed particles similar to the in vitro-incorporated influenza; these particles were not observed in control platelets (Fig. 3c, Supplementary Fig. 8). Our findings indicate that influenza virus during acute influenza infection can potentially cross into blood and become engulfed by platelets.

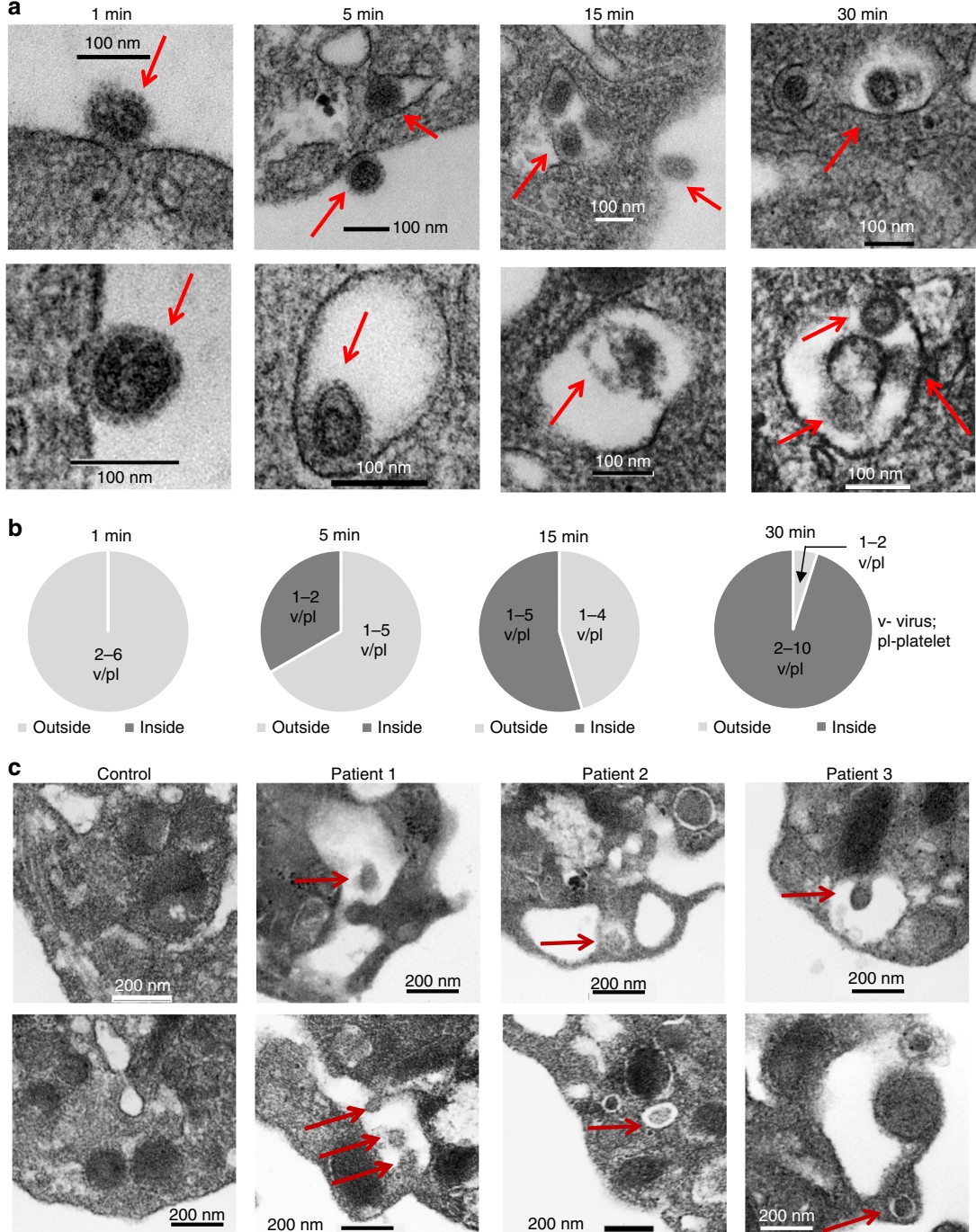

**Fig. 3** Transmission electron microscopy of influenza particles in platelets. **a** Transmission electron microscopy (TEM) of isolated human platelets incubated with WSN/33 influenza (1 pfu to 10 platelets) at different time points but under the same conditions as in (**b**). Platelets were fixed for 10 min immediately after incubation and processed for electron microscopy. **b** Quantitation of the time course of internalization of influenza by isolated platelets in (**c**). The graphs represent the analysis of $n = 10$ platelets per time point. **c** TEM of platelets isolated from uninfected (control) and influenza-infected patients. Representative images are shown from three different influenza patients. In all cases, red arrows point toward the viral particle

**Platelets internalize influenza possibly by phagocytosis.** Phagocytosis is the process by which blood cells clear pathogens such as influenza. Using transmission electron microscopy and in vitro incubation of platelets with influenza over time, we found morphology consistent with the classical stages of phagocytosis (Fig. 4a). Negative staining of the viral particles can be visualized in Fig. 4b. Attachment was observed as early as 1 min post-incubation followed by invagination and formation of phagosome-like structures (PLS) (Fig. 4a). In select images, there was fusion with platelet granules that do not appear to be alpha granules but may be of lysosomal origin. Digestion of viral particles and residual body formations can also be visualized in the PLS. The capacity of platelets to phagocytose has been controversial and with certain pathogens such as bacteria, it has been proposed that they are covercytes rather than phagocytes[21]. The covercyte ability comes from the claim that uptake of bacteria

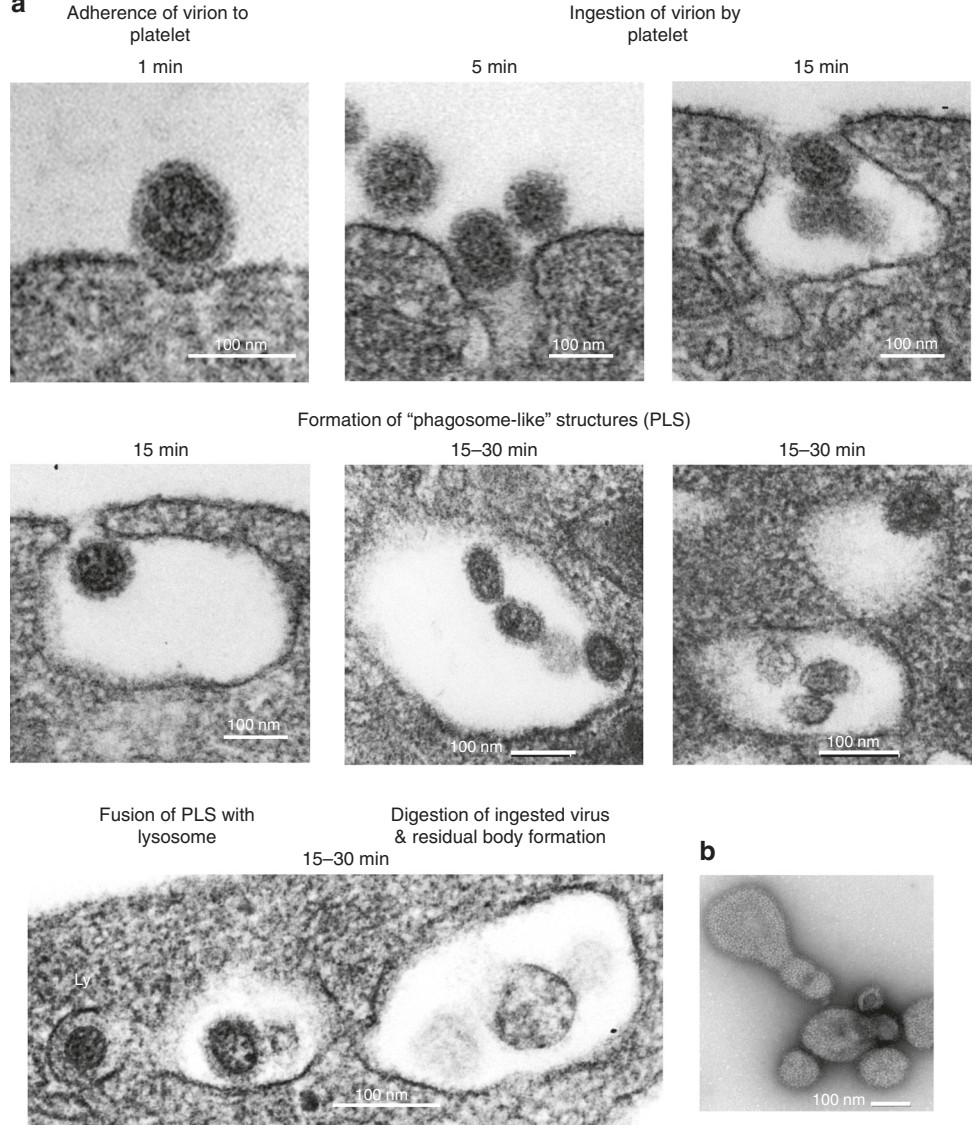

**Fig. 4** Stages of phagocytosis in platelets' internalization of influenza. Phagocytic morphological features of influenza internalization by platelets was assessed by TEM. **a** Representative images of different stages of phagosome-like structures assessed by morphological features captured in platelets from healthy human donors incubated with WSN/33 influenza (1 pfu to 10 platelets) at different time points. Five distinct stages can be observed in human platelets. **b** TEM of the negative stain of influenza virus (only)

involves channels of the open canalicular system (OCS) which would not be involved in phagocytosis. We found that PLS did not have an apparent connection to the OCS (Fig. 4a). The morphology of influenza engulfment by platelets appears to recapitulate all major stages of phagocytosis by forming vacuoles without the involvement of the OCS, and this suggests that for viral particle removal, platelets may have the capacity to act as phagocytes.

**Influenza leads to TLR7-dependent C3 release from platelets.** TLR7 is a major pattern recognition receptor involved in the initial recognition of engulfed influenza likely by sensing viral ssRNA content[17]. Platelets express functional TLR7 and TLR7 stimulation leads to alpha granule protein release[6] while neutrophils do not express TLR7 at the protein level[22]. Lack of expression of TLR7-mRNA in platelets, however, has been observed in some human donors[11]. The complement system is also part of the innate immune response and platelets contain C3

in their granules[23,24]. We first screened plasma isolated from the influenza-infected patients to assess if C3 is released during influenza infection in vivo and we found increased levels of circulating C3 compared to control (Fig. 5a). Interestingly, aspirin intake in patients did not have an effect on C3 plasma levels, suggesting a COX-independent mechanism for C3 release (Supplementary Fig. 9a). We then examined influenza–platelet interactions to assess direct contribution of platelets to C3 release as a function of TLR7 activation. We isolated human platelets from control donors and incubated them with infectious virions of influenza A (WSN/33) for 30 min at three different ratios to establish TLR7-specificity: 1 plaque-forming unit (pfu) or infectious virion to 10, 100, or 1000 platelets. The presence of influenza led to an increase of C3 in platelet supernatants (Fig. 5b) but only in those with platelet TLR7 expression as assessed by qPCR or antibody staining. Inhibition of TLR7 with a specific TLR7 antagonist, IRS661, abrogated the release of C3 from platelets (Fig. 5b). Additionally, C3 release from platelets in 30 min was

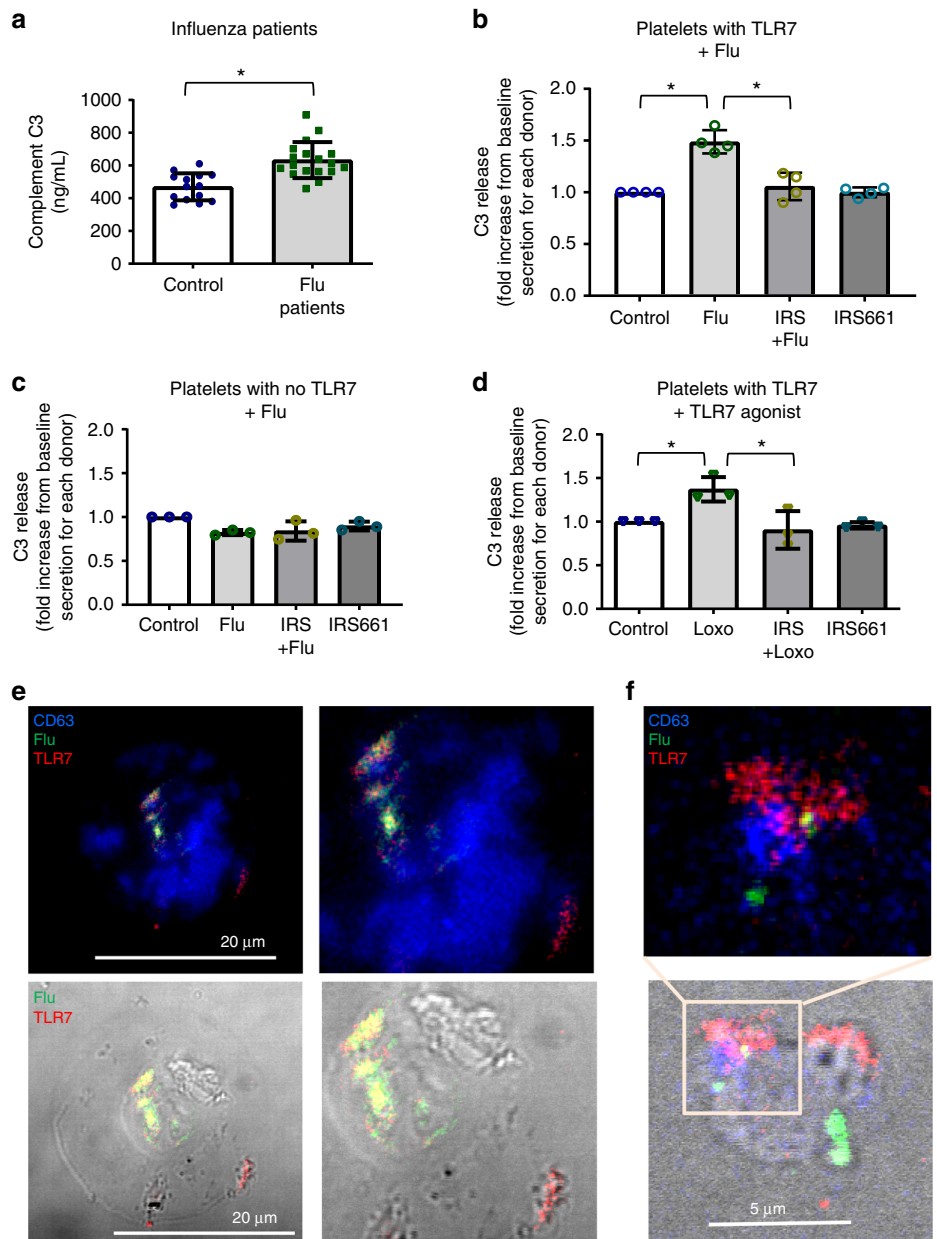

**Fig. 5** Influenza mediates C3 release from platelets through TLR7. **a** C3 in plasma from healthy donors ($n = 14$) and influenza-infected patients ($n = 18$). The graphs represent the average ± SD; significance was assessed by Mann–Whitney $U$ test, *$p < 0.0001$. **b** C3 release from human platelets that express TLR7 (assessed by qPCR) had been isolated from healthy donors and mixed with influenza strain WSN/33 at a proportion of 1 pfu to 100 platelets ($n = 4$, 3F, 1M); $p < 0.0001$, $F = 27.49$, df = 3. **c** C3 release from human platelets that do not express TLR7 (assessed by qPCR) treated as in (**b**). Graph is representative of $n = 3$ different blood draws; $p = 0.2813$, $F = 1.833$, df = 3. **d** C3 release from human platelets from the same donors as in (**b**) treated with the TLR7 agonist loxoribine 1 mM in the presence or absence of TLR7 inhibitor IRS661; $p = 0.0086$, $F = 7.989$, df = 3. **b–d** Data in graphs are represented as average ± SD; significance was assessed using ANOVA followed by Bonferroni multiple comparison test and star symbol (*) indicates $p < 0.05$. Source data for (**a–d**) are provided as a Source Data file. **e** Confocal images of permeabilized isolated platelets from influenza-infected patient stained for flu-FITC; TLR7-APC; lysosomal marker CD63-BV421. **f** Confocal images of isolated healthy human platelets incubated with WSN/33 for 30 min (1 pfu to 10 platelets) and stained as in (**e**). **e**, **f** Representative images of platelets from $n = 4$ (2M, 2F) different donors are shown

observed only when we incubated 1 pfu of influenza with 100 platelets; C3 release using a high proportion of virions to platelets was not specific for TLR7; and the low proportion did not elevate detectable C3 in the supernatants after 30 min of incubation (Fig. 5b). Isolated platelets that did not express TLR7 had no detectable increase in C3 release (Fig. 5c). This limited observation suggests that polymorphisms of TLR7 could also influence the outcome of infection in certain population groups[25]

(see Supplementary Note 1). Similar to influenza-stimulated platelets, stimulation of isolated human platelets with the TLR7 agonist, loxoribine (Loxo), led to an increase in C3 release from platelets and antagonism of the receptor abrogated the effect (Fig. 5d, Supplementary Fig. 9b, c).

TLR7 is located in vesicles of lysosomal origin and it is a dual receptor for guanosine and uridine-rich single-stranded RNA[26]. As a result, it does not recognize proteins of the viral particle but

rather senses influenza nucleic acids. To assess if influenza and TLR7 can be located in the same platelet, we isolated platelets from an influenza-infected donor and stained for influenza, TLR7, and the lysosomal marker CD63. All of the markers colocalized or were found in close proximity to each other (Fig. 5e). Similar results were observed in platelets incubated with influenza in vitro (Fig. 5f). Interestingly, TLR7 was not always found colocalized with the lysosomal marker CD63 or LAMP1 (Fig. 5f or Supplementary Figs. 10, 11a). Additionally, not all platelets that contained viral particles expressed TLR7 (Supplementary Fig. 11). Platelets however are known to take up RNA from endothelial cells or leukocytes[5,27,28], therefore, transfer of RNA between other cell types after viral digestion is a valid possibility. Scanning and transmission electron micrographs of platelets incubated with influenza showed interaction and pore formation in platelets as well as disappearance of the cell wall between them (Supplementary Figs. 12, 13). In total, our data suggests that engulfed influenza nucleic acids are sensed by platelet-TLR7 and this sensing leads to complement C3 release.

**C3 and GM-CSF effect on neutrophil-DNA release**. Since granulocyte-macrophage colony-stimulating factor (GM-CSF) primes neutrophil TLR responses and delays spontaneous neutrophil death[29–32] we evaluated if platelet-TLR7 signaling is involved in GM-CSF secretion. Interestingly, platelets secreted GM-CSF only in the presence of neutrophils as a function of TLR7 stimulation; platelet-TLR7 without neutrophils had no effect on GM-CSF levels (Fig. 6a–c).

It is not currently known if C3 mediates release of DNA from neutrophils, particularly in circulating unattached cells. Therefore, we treated human neutrophils (in suspension and under constant rotation) with native C3. C3 alone initiated DNA release and neutrophils formed very large aggregates (Fig. 6d, e). Incubation of platelet-free neutrophils with C3 in the presence of GM-CSF, however, reduced the size of the neutrophil aggregate formations (Fig. 6d, e). These data suggest that influenza causes C3 release from platelets, and that platelet-TLR7 is involved in this process. Our results also suggest that C3 alone is sufficient to mediate DNA release from neutrophils and platelet-GM-CSF controls levels of this release. This provides a previously unrecognized link between TLR7 and complement during viral infection.

**Platelet-TLR7-dependent C3 induces neutrophil-DNA release**. As noted above, we observed free aggregates of released neutrophil-DNA and platelets in the blood of influenza-infected patients. To evaluate if TLR7 contributes to release of neutrophil-DNA directly in blood (without neutrophil adherence to the endothelium) we treated blood from healthy donors with a TLR7 agonist and kept the blood suspended by incubating at constant rotation for 30 min at 37 °C (using an aggregometer). Confocal microscopy of the TLR7 agonist-stimulated blood showed an increased release of neutrophil-DNA with attached platelets as compared to control blood (Supplementary Fig. 13). These neutrophil-derived DNA-platelet aggregates were sometimes associated with MPO and histones consistent with NET formation (Supplementary Fig. 13).

To understand the specific contribution of TLR7 signaling, we separately isolated platelets and neutrophils from human blood using a slightly modified method to generate a neutrophil population that rarely contained platelets (Supplementary Fig. 14). Neutrophils or platelets were pretreated for 15 min with a TLR7 agonist and then the two populations were mixed and incubated together for 30 min. Utilizing this method, we demonstrated that platelet-TLR7 (but not neutrophil-TLR7)

mediated neutrophil-DNA release independent of attachment (Fig. 7a, b). In the supernatants of these mixing experiments, however, we were unable to detect suggested markers of NETosis[33] such as cell-free double stranded DNA (by using PicoGreen®), citrullinated histone H3, or free histone H4 in the supernatants (all values were below detection limit). These data suggest that influenza-mediated neutrophil-DNA released in the circulation may not be characterized by the same markers of NETosis that are observed when neutrophils are attached and platelet-free[20]. Additionally, our data indicate that the process of influenza-mediated, intravascular (vital) NETosis (without endothelial attachment) is tightly regulated by platelets.

To establish if the platelet-TLR7-C3 axis mediates the release of DNA from neutrophils, we pretreated the co-incubated isolated platelets and neutrophils with compstatin, an inhibitor of C3, and then treated with a TLR7 agonist. Treatment with the C3 inhibitor significantly reduced neutrophil-DNA release driven by TLR7 activation (Fig. 7c). To further evaluate the contribution of the TLR7-C3 axis to neutrophil-DNA release, we co-incubated isolated platelets and neutrophils together and treated them with influenza in the presence and absence of a TLR7 antagonist. The inhibition of TLR7 reduced influenza-mediated DNA release from neutrophils (Fig. 7d) but not to the full extent (Fig. 7e). These data indicate that the effect of C3 on neutrophil-DNA release is mediated by TLR7 but the effect of influenza on neutrophil-DNA release may not be solely dependent on TLR7.

**Platelets mediate myeloperoxidase release from neutrophils**. MPO is a major enzyme that deposits along released neutrophil-DNA during NETosis. Additionally MPO in plasma has been linked to inflammation and increased risk for acute MI[34]. To assess if platelet-TLR7 can contribute to MPO levels, we stimulated neutrophils in the presence or absence of platelets. Interestingly, following 30 min of TLR7 stimulation, MPO release from neutrophils was elevated only when platelets were present, and no significant changes in MPO were observed in the neutrophil fractions without platelets (Fig. 8a, b). Of note, in select individuals, neutrophils have attached platelets that could contribute to the non-significant increase of MPO with agonists seen in Fig. 8a. TLR7 stimulation of neutrophils in the presence or absence of platelets for 30 min did not lead to detectable release of IL-8, citrullinated H3, or CCL5 (all values were below values for incubation media); CCL5 changes were not detected in plasma of influenza patients as compared to control (Supplementary Fig. 15). IL-8 mRNA has been found in human platelets[27] and given its role in neutrophil chemotaxis, we evaluated its possible secretion as a function of TLR7. These data suggest that in the initial response to pathogens and early stages of the TLR-mediated response platelet presence stimulated the release of MPO from neutrophils without affecting primary interferon response or neutrophil chemotaxis.

**Ly6G-DNA release is platelet- and TLR7-mediated in mice**. Blood from patients with acute influenza cannot be well controlled with respect to synchronized development of influenza infection. For that purpose, we utilized a murine model to assess specificity and platelet contribution to C3 and DNA release from bright Ly6G cells that are mostly neutrophils (but may include some eosinophils) in the circulation of mice. Of note, we have previously shown that with respect to neutrophil engagement, TLR7 in platelets of mice function in a similar manner as in humans[6]. To address TLR7-specificity for the DNA release from Ly6G-positive cells in a controlled and time-dependent manner, we challenged C57BL/6J wild type (WT) and TLR7 knockout

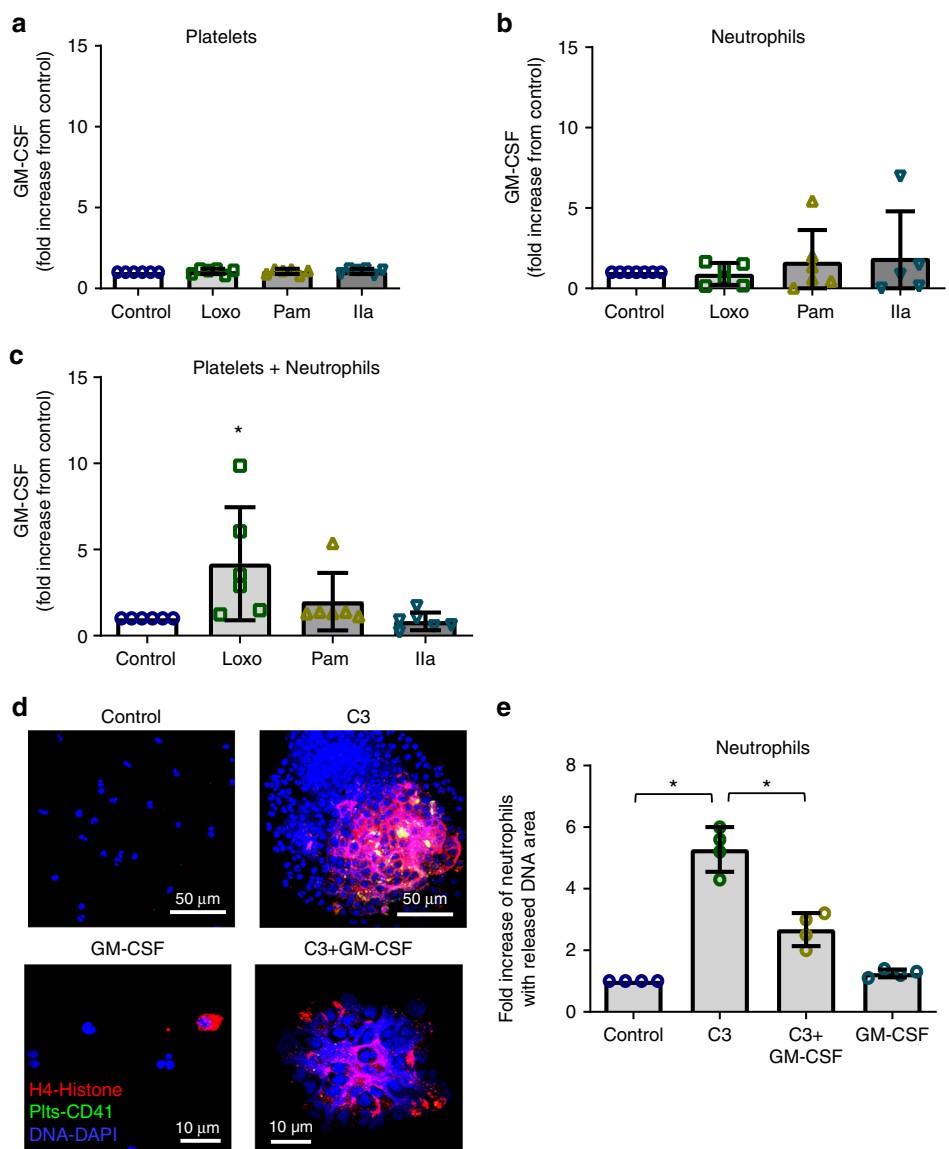

**Fig. 6** Role of platelet GM-CSF and C3 in neutrophil-DNA release. **a–c** Isolated human platelets and neutrophils were incubated together or by themselves for 30 min (at constant rotation and 37 °C) in the presence of TLR agonists [TLR7 (Loxo)—1 mM; TLR2 (Pam$_3$CSK$_4$, PAM)—10 µg/µL] or thrombin (IIa)— 0.05 U/mL. GM-CSF release from **a** platelets ($p = 0.8903$, $F = 0.2071$, df = 3), **b** neutrophils ($p = 0.7255$, $F = 0.4422$, df = 3), and **c** platelets and neutrophils incubated together was measured by ELISA ($p = 0.0200$, $F = 4.116$, df = 3). The graphs represent the average fold change for each individual of $n = 6$ (3F; 3M) ± SD. **d** Confocal images of isolated human neutrophils treated with C3 (30 ng/mL) and neutrophils treated with C3 in the presence of GM-CSF (25 ng/mL). Neutrophils were treated in HEPES-modified Tyrode's buffer ($0.04 \times 10^5$ neutrophils/µL) for 30 min at 37 °C, and constant rotation (aggregometer). At the end, cells were fixed and stained [CD41-FITC-platelets (green); H4-AF637-histone (red); DAPI-DNA (blue)]. Neutrophil-DNA aggregates were visualized by confocal microscopy. Images are representatives of $n = 4$ different donors. C3-treated neutrophils from healthy donors release their DNA and form large aggregates. **e** Quantitation of (**d**). The graph ($n = 4$, 2F, 2M) is represented as average ($p < 0.0001$, $F = 71.42$, df = 3) ± SD. Significance in (**a–c**, **e**) was assessed using ANOVA followed by Bonferroni multiple comparison test and star symbol (*) indicates $p < 0.05$. Source data are provided as a Source Data file

(KO) mice with a TLR7 agonist (loxoribine) or influenza virus. Released Ly6G-positive DNA was evaluated in fixed blood immediately after collection by cardiac puncture and samples were examined by confocal microscopy. Mice challenged with loxoribine (Fig. 9a, b) or influenza (Fig. 9c, d) exhibited observable DNA presence in the circulation 24 h after challenge; 4 h were not sufficient for detectable differences in DNA release (Supplementary Fig. 16). In vivo DNA release from Ly6G-expressing cells (Fig. 8, Supplementary Fig. 17) was found to be TLR7-specific, as the TLR7-stimulation did not lead to observable

DNA release in TLR7 KO mice. Of note, in our hands staining of blood did not show positive expression of Ly6G in lymphocytes or monocytes (Supplementary Fig. 17). Similar to human blood, the released Ly6G-positive-DNA was found in the circulating blood unattached and was rarely free from platelets. Additionally, influenza virions were found in the plasma of TLR7 KO mice (Supplementary Fig. 17) implying possible necessity for TLR7 activation during the initial stages of infection. Our data suggest that endothelium-free Ly6G-positive-DNA can be found in the circulation in a TLR7-specific manner.

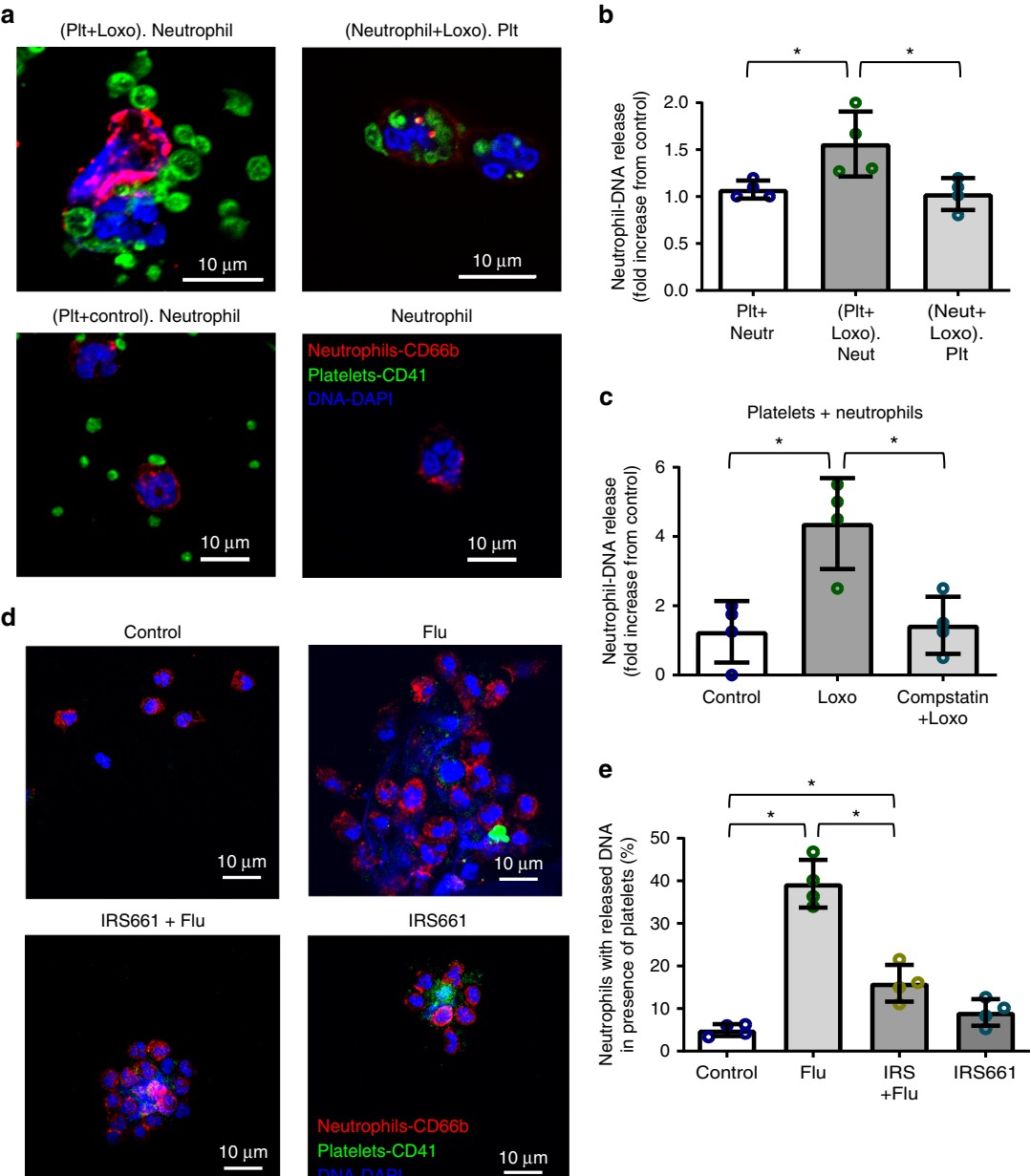

**Fig. 7** Platelet-TLR7 solely mediates neutrophil-DNA release through C3. Platelets and neutrophils were isolated from human blood. Each population was pretreated for 15 min with a TLR7 agonist (1 mM Loxoribine, Loxo) and then incubated with the other population for 30 min. All experiments were carried out at an approximately physiological ratio of 50 platelets:1 neutrophil. **a** Confocal microscopy of the incubated cells. Cells were stained with (CD41-FITC-platelets; CD66b-APC-neutrophils; DAPI-DNA) and visualized with a confocal microscope. **b** Quantitation of the confocal images in (**a**), $p = 0.017$, $F = 6.624$; df = 2. **c** DNA release from neutrophils in the presence of platelets pretreated with a C3 inhibitor, compstatin (0.088 mg/mL) for 10 min and then stimulated with Loxo for 30 min ($p = 0.0033$, $F = 11.51$, df = 2). **d** Assessment of neutrophil-DNA release by confocal images of isolated platelets and neutrophils treated with influenza WSN/33 in the presence or absence of the TLR7 inhibitor IRS661. **e** Quantitation of the confocal images in (**d**), $p < 0.0001$; $F = 61.07$, df = 3. In all cases data in the graphs are represented as the average ± SD. Statistical significance was measured by ANOVA followed by a Bonferroni follow-up test of $n = 4$ (2F and 2M, with the exception of (**e**), where we used 3F and 1M), star symbol (*) indicates $p < 0.05$. Source data for all graphs are provided as a Source Data file

In order to assess the contribution of platelets to Ly6G-positive-DNA release and C3 secretion in vivo, we ablated platelets with anti-CD42 antibody as we have previously reported[6]. At 24 h post-ablation, when the platelet level decreases to 0.01% (as assessed by blood count analyzer), we infected mice with influenza virus. Ablation of platelets at the beginning of infection led to reduced release of Ly6G-positive-DNA aggregates in blood (Fig. 10a, b). Consistently, C3 in plasma was also reduced in the mice with ablated platelets compared to control infected mice (Fig. 10c). Isolated platelets from the initially platelet-ablated mice showed a pattern of more viral RNA 12 days post-infection than mice injected only with IgG (Fig. 10d). This difference however was not statistically significant assuming Gaussian distribution (Fig. 10d); Mann–Whitney nonparametric test, $p = 0.0286$. Our data suggest that platelets contribute to C3 secretion and mediate DNA release from Ly6G-positive cells

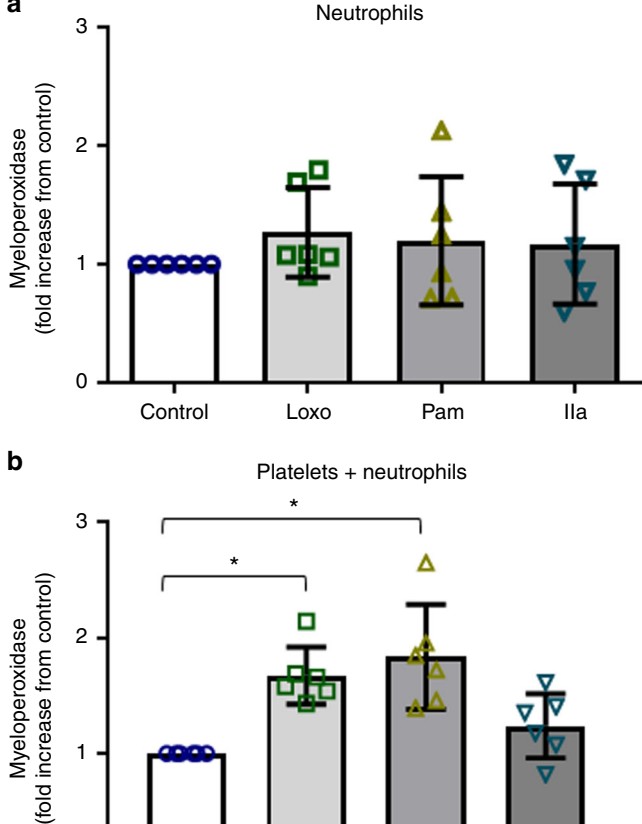

**Fig. 8** Platelet-TLR7 mediates release of MPO from neutrophils. Isolated human platelets and neutrophils were incubated together or by themselves for 30 min, at 37 °C and constant rotation, in the presence of TLR agonists [TLR7 (Loxo)—1 mM; TLR2 (Pam)—10 μg/mL] or thrombin (IIa)—0.05 U/ mL. Myeloperoxidase (MPO) release from neutrophils was measured in the **a** absence ($p = 0.7206$, $F = 0.4493$, df = 3) and **b** presence of platelets by ELISA ($p = 0.0002$, $F = 10.4$, df = 3). The graphs represent the average fold change for each individual of $n = 6$ (3F; 3M) ± SD. Source data are provided as a Source Data file. Significance was measured by ANOVA followed by Bonferroni follow-up test; in all cases star symbol (*) indicates $p < 0.05$

(mostly neutrophils but may include some eosinophils) during influenza infection.

## Discussion

While the initial host response to intravascular pathogens is traditionally believed to be primarily controlled by leukocytes, particularly neutrophils, more recent data have demonstrated that this response may also require platelets[5,16]. In this study, we show that influenza is found in the circulation where it is engulfed and recognized by platelet-TLR7 leading to complement C3 release. Platelet-C3, in turn, stimulates neutrophils to release their DNA, resulting in a potentially highly prothrombotic process that may contribute to elevated risk of MI. Individually, elevated levels of complement C3 and coronary NET burden are predictors of acute MI and myocardial infarct size, respectively[14,18,35]. Our study connects these independent contributors to MI and suggests a potential mechanism. We further demonstrate that the release of neutrophil-DNA directly in blood requires platelets but does not require attachment to the endothelium. We demonstrate

an increase in C3 in influenza-infected patients' plasma. We outline a novel mechanism by which platelets, through TLR7, modulate the increase in C3 release, providing a previously undescribed link between these two innate immune pathways. In the presence of TLR7-activated platelets, neutrophils provide a signal that leads to the secretion of GM-CSF from platelets. GM-CSF reduces the level of C3-mediated neutrophil-DNA release establishing a feedback mechanism. Lastly, neutrophil anti-microbial potential, as measured by release of MPO, appears to be mediated by platelet-TLR7 and is independent of thrombin stimulation (Fig. 10e). Thus, we conclude that platelets contribute to the complex response of neutrophils to pathogens, particularly in the setting of influenza.

Viremia during influenza in humans is not a uniform observation[10,36]. Fluorescent staining of blood platelets from influenza-infected patients analyzed in this study show that platelets have viral particles inside of them. Transmission electron micrographs of platelets incubated in vitro with purified infectious influenza show that the virus is rapidly internalized by platelets and the internalization process is morphologically consistent with phagocytosis. As mentioned, phagocytic activity of platelets is controversial; however, we were able to observe all major stages of phagocytosis in platelets. Future studies are necessary to assess if this process is typical only of small particles such as viruses and whether it occurs with bacteria. Additionally, the internalization of influenza by platelets is a rapid process and by 30 min, in some platelets, there may be up to 10 viruses with viral particles distributed throughout different vacuoles. However, it is important to stress that we do not have evidence that platelets, at any point, become infected with influenza. As platelets have no nucleus in which viral replication can occur, our findings suggest only that platelets sequester influenza from the circulation, digest the virus, and thereby lead to activation of the innate immune system. Perhaps the extensive number of platelets provides an increased surface area during viral infection to support phagocytosis and remove leaked viral particles from the circulation. Thus, the numerous, anucleate nature of platelets may be an evolutionary advantage supporting decreased viral reproduction and increased communication with other immune cells. Overall, platelets may be the first intravascular defense mechanism.

Influenza infection in humans has been noted to lead to changes in platelet reactivity[7], suggesting that platelets may contribute to the prothrombotic/inflammatory response during infection. However, in humans, reduced platelet count as a consequence of influenza infection is inversely proportional to mortality risk and acute respiratory failure[37]. Furthermore, the satellite platelet morphology (in addition to thrombocytopenia during influenza) may also represent a dysfunctional platelet population that may not be able to support proper alveolar endothelial barrier in the capillary of the lungs[25] and some viral particles may leak into the circulation. Additionally, dysfunctional platelets may not be able to interact properly with released DNA in the circulation as evidenced by the spread-out interaction between platelets and the DNA in our fluorescent images. In vitro studies of human platelets incubated with heat-inactivated influenza virions have shown that influenza A can activate platelets through the FcγRIIa receptor leading to thrombin secretion[38]. Studies of patients with complications from influenza infection have implicated damage to cardiac[39,40] and skeletal muscle[41] as well as the brain[7,42] and liver[7,43], although the organs involved rarely contained infectious virus. The mechanism by which influenza virus reaches distant tissues outside the respiratory tract and achieves systemic inflammation and organ damage is still poorly understood. Our findings propose that dysregulated platelets, or thrombocytopenia, may contribute to the transport of viral particles to distant tissues.

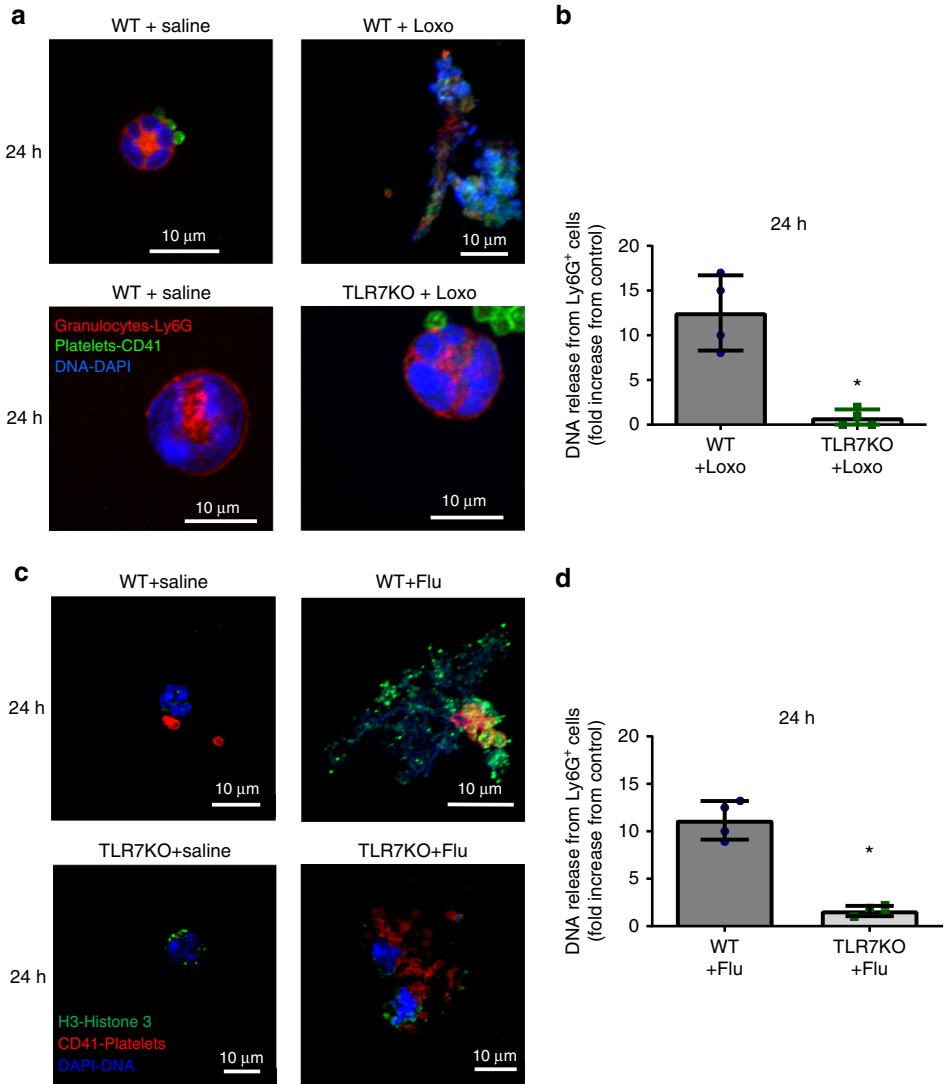

**Fig. 9** TLR7 stimulation in vivo leads to DNA release from Ly6G positive cells. WT and TLR7 KO mice were injected intraperitoneally with a TLR7 agonist, or were intranasally infected with influenza (PR8 strain, 40,000 pfu in 30 μL). Blood was collected by cardiac puncture at 24 h and immediately fixed (red blood cells were lysed at the same time); Ly6G is predominantly expressed by murine neutrophils. **a** Representative images of DNA release from Ly6G-positive cells (at 24 h post-Loxo stimulation) resolved by confocal microscopy. **b** Quantitation of DNA release from Ly6G-positive cells in blood of mice (n = 4/group) at 24 h after agonist stimulation (p = 0.016, df = 6). **c** Representative images of DNA release (at 24 h post influenza infection) resolved by confocal microscopy. Pictures showing Ly6G-highly positive origin of the released DNA are included in Supplementary Fig. 11. **d** Quantitation of the DNA release from Ly6G-positive cells in blood of mice (n = 4/group) at 24 h post-infection (p < 0.001, df = 6). In all cases, the bar represents 10 μm and values in the bar graphs represent the average ± SD; star symbol (*) indicates p < 0.05. Significance was assessed by unpaired t-test (two-tail value). Source data are provided as a Source Data file

The complement system is an important component of the innate immune response involving pathogen opsonization, chemotaxis of inflammatory cells, and lysis of infected cells, ultimately removing immune complexes, apoptotic cells, and cell debris. Complement activation is increasingly recognized as a major contributor to cardiovascular disease and intravascular inflammation[40]. Human platelets contain C3, complement C4 (C4) precursor, and complement C1 inhibitor in their alpha granules[8]. Human neutrophils, on the other hand, do not contain complement components but have one positive regulator of complement activation (properdin)[36]. Interestingly, C3 (and C4) deposition on the viral envelope of influenza virus (WSN/33) is known to activate the complement system and leads to viral neutralization[44]. Here, we report a novel mechanism by which platelets cross-communicate with neutrophils to mediate the initial intravascular response to influenza infection through C3. As the infection progresses increased presence of C3 in plasma, coming from the liver or from dysregulated platelets, may overwhelm its beneficial effect. C3-mediated circulating aggregates could explain the increased risk of cardiovascular events in select patients. During intravascular bacterial infection, platelets are necessary for building proper antimicrobial immunity utilizing GP1b and C3[10]. Interestingly, during sepsis, C3 proteolysis is predictive of the severity of infection and sepsis can result in MI[45,46]. Perhaps circulating neutrophil–DNA–platelet aggregates, although necessary for viral (or bacterial) removal and adaptive immunity during infection, may become pathological throughout the course of infection with increased vascular damage and possible MI burden.

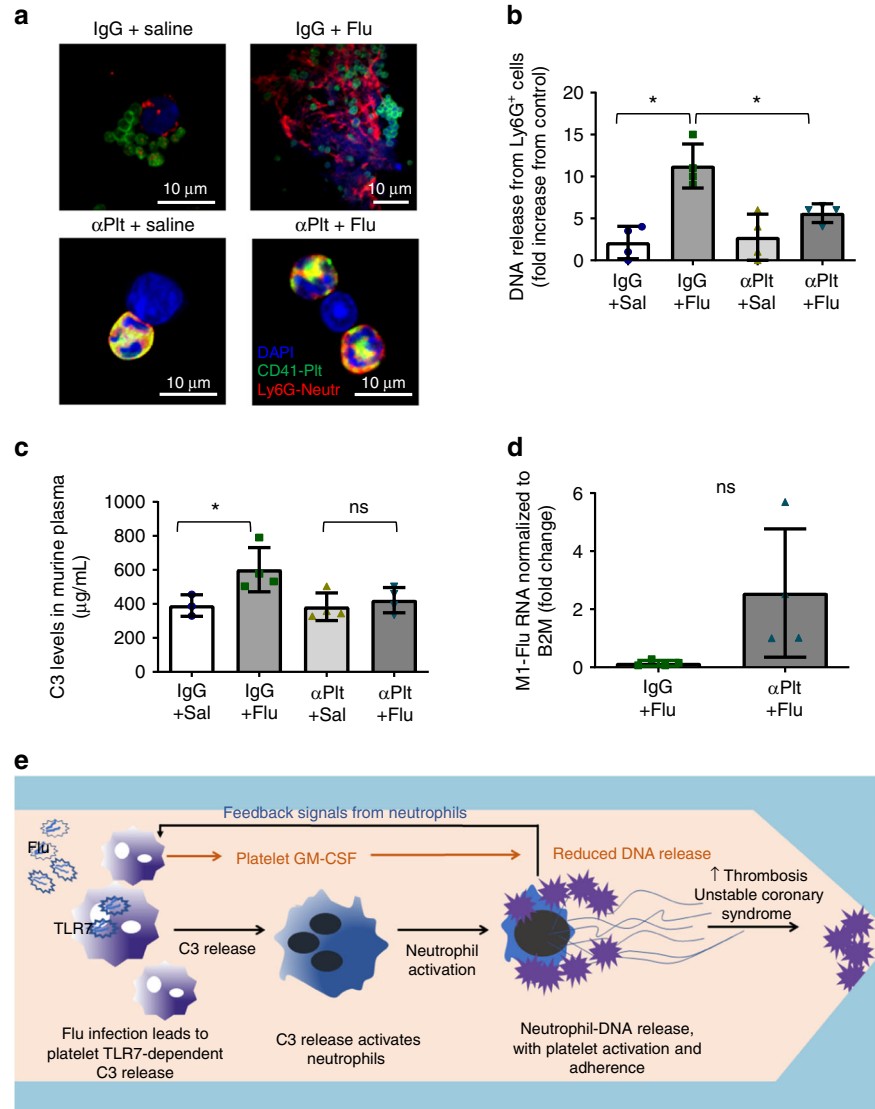

**Fig. 10** Platelets contribute to C3 and Ly6G-DNA release in vivo. Platelets were eliminated from male mice with antiplatelet antibody CD42 (αPlt) and compared to control IgG. At 24 h post elimination, mice were infected with the PR8 strain of influenza (as in Fig. 8). **a** Confocal microscopy of blood showing DNA release in murine blood 3–4 days post-infection. Images are representative of $n = 4$ mice/group. **b** Quantitation of the confocal images in (**a**). Graph is a representative of 4 mice/group ($p = 0.0003$, $F = 14.26$, df = 3). **c** C3 levels in murine plasma at the same time as in (**a**). The graph represents the average levels ± SD, $n = 4$ mice/group, with the exception of IgG+sal, where $n = 3$ mice were used ($p = 0.0415$, $F = 4.733$, df = 3). Significance was measured by ANOVA followed by Bonferroni follow-up test; in all cases star symbol (*) indicates $p < 0.05$. **d** Gene expression levels of influenza RNA in isolated murine platelets 12 days post-infection ($n = 4$ of IgG+flu; $n = 4$ of αPlt+flu). The graph represents average expression ± SD; significance was calculated by two-tailed unpaired $t$-test, $p = 0.0715$, df = 6. Of note, Mann–Whitney non-parametric $t$-test gave $p = 0.0286$. Source data are provided as a Source Data file. Abbreviations: IgG—control antibody; αPlt—antiplatelet CD42b antibody; Sal—phosphate buffered saline: **e** Proposed mechanism of platelet-mediated neutrophil-DNA release during influenza infection. During influenza infection, virions cross into the circulation and become engulfed by platelets. Influenza virions lead to the release of complement C3 from platelets in a platelet-TLR7-dependent manner. C3 in turn activates neutrophils to release their DNA and leads to the formation of platelet–neutrophil aggregates that can circulate freely in blood. Aggregates of this nature can increase the risk for thrombosis and potentially lead to unstable coronary syndrome when there is vessel stenosis or inflamed endothelium

Neutrophil extracellular traps (NETs) can cause host tissue damage and neutrophils with increased netting potential are seen in inflammatory or autoimmune diseases such as diabetes, systemic lupus erythematosus, and rheumatoid arthritis[47–50]. Platelets have been associated with organ NETosis in vivo, but this has been thought to be mostly a reactive process[16,20,51]. Here, we report that platelet-TLR7 solely mediates the release of DNA from neutrophils directly in the circulation without the requirement of attachment. These observations, in addition to the lack of

TLR7-protein in human neutrophils, suggest that platelets are important contributors to neutrophil activation during certain viral infections. We also observed that activation of platelet-TLR7 also leads to feedback signaling from neutrophils that consequently initiates GM-CSF release from platelets. GM-CSF, in turn, is known to have anti-apoptotic potential and extends neutrophil survival[30]. Differential secretion of GM-CSF with TLR7 agonists but not with TLR2 agonists suggests a sophisticated platelet–neutrophil cross-communication mechanism that

controls the intensity of neutrophil-DNA release in blood depending on the type of pathogen (viral or bacterial). We postulate that during influenza infection, platelets are not only able to initiate DNA release from unattached neutrophils but can control the amount of released DNA by secreting GM-CSF and possibly by providing a physical barrier between neutrophils. This particular function may be necessary to capture substances circulating in blood, activate the immune system but may lead to unwanted MI risk.

In addition to the pro-inflammatory and autoimmune contribution, a pathological outcome of NETosis is thrombosis. In vitro studies treating purified neutrophils with PMA have shown that NETosis and thrombosis may be concordant[52]. Histones released during NETosis may increase plasma thrombin generation in a platelet-TLR2, TLR4-dependent manner[53,54]. In fact, extracellular histones during sepsis are a major contributor to death, and infusion of histones in mice leads to the formation of platelet-rich microthrombi and consequent thrombocytopenia[55]. In our in vitro cell mixing experiments, we were unable to detect free citrullinated histones after stimulation with TLR7 agonist for 30 min. This is not surprising given reports of cases in which histone citrullination is not a marker of NETosis[19,33]. Of note, the observed neutrophil-DNA release in this study was surrounded by platelets, a process that may impair the detection of histones, nucleosomes, or free DNA. Additionally, the result of this DNA-release is the formation of large platelet-DNA aggregates that in most cases showed traditional markers of NETosis; however, in the influenza-infected patients we cannot exclude the contribution of other cells to this process. It is important to mention that NETosis during influenza infection differs from suicidal-NETosis (PMA-mediated) as it is PAD4-independent and negative for citrullinated histone H3[56]. Regardless of the process, the uncontrolled formation of platelet-DNA aggregates ultimately becomes problematic for influenza patients in the first 7 days post infection and increases the risk for MI. Elucidating the mechanisms of platelet–neutrophil interactions in the initial stages of infection is critical in understanding the processes that may contribute to uncontrolled immune response or thrombosis. Additionally, understanding initiation and mediation of neutrophil-DNA release directly in blood is particularly important since this neutrophil function relates to cardiovascular disease and NETs are presumed to be highly prothrombotic[57,58], are found in deep vein thrombosis and are associated with atherosclerosis[57,59,60].

In this study, we focused on understanding how influenza infection in humans may lead to an increased risk for acute MI. Murine models utilized herein provided controlled in vivo tools for assessing specificity and contribution of platelets to neutrophil-DNA release and complement secretion in a time-specific manner. Although, we conclude unequivocally that platelets are important in both species, unstable coronary syndromes do not occur in non-genetically manipulated mice and mice are not naturally infected by influenza. Also, with respect to the two species and influenza infection, the bronchial–epithelial response is markedly different the proportions of platelets to neutrophils in blood that increases from 50:1 in humans to at least 500:1 in mice, and the most prevalent leukocyte in human blood is the neutrophil, whereas in mice, it is the lymphocyte. Murine studies have shown that elimination of platelets from mice before infection with influenza leads to increased survival due to a profound thrombotic response in lungs[61]. Thrombocytopenia severity in humans however predicts mortality during influenza infection. Differences between mouse and human platelets have been described previously[62]. The observations in human patients suggest that, during influenza infection, platelets may play an important role in human blood that may not be fully predicted by the behavior of murine platelets.

Future studies may focus on understanding the contribution of other infections to platelet activation and neutrophil-DNA release and generality of this mechanism. It will also be necessary to understand C3 regulation as a function of non-pathogenic inflammation and to evaluate if regulation of C3 by inhibitors such as compstatin could attenuate cardiothrombotic events and reduce the risk of influenza-mediated MI.

Our study demonstrates that, during influenza infection, platelets internalize influenza and coordinate a distinct, multifactorial response as a function of their TLR7-mediated C3 release. Platelets appear to be major contributors to the initial response to pathogens by bridging the TLR and complement systems through C3 secretion and by mediating MPO secretion and release of DNA from neutrophils. TLR7-activated platelets also secrete GM-CSF but only when neutrophils are present. GM-CSF, which is known to promote neutrophil survival, in turn, reduces the amount of released DNA. Our findings suggest that the initial immune response to influenza infection requires intravascular communication between platelets and neutrophils. Dysregulation of this response and dysfunctional platelets may act as a double-edged sword and may lead to increased MI risk in select individuals.

## Methods

**Pharmacological compounds**. This study used the following compounds: loxoribine (InvivoGen, CA, USA, cat# tlrl-lox), Pam$_3$CSK$_4$ (InvivoGen, CA, USA, cat# tlrl-pms), human thrombin (Enzyme Research Laboratories, IN, USA, cat# HIIa), prostaglandin E1 (PGE1, Millipore, MA, USA, cat# 538903-1MG), complement C3 (Millipore, cat #204885); GM-CSF (Stemcell Technologies, MA, USA, cat #78015.1), and compstatin (Tocris, MN, USA, cat# 2585). Thrombin and Pam$_3$CSK$_4$ were dissolved in water and blood or isolated cells were treated with 10 μg/mL of Pam$_3$CSK$_4$ or 0.05 U/mL of thrombin. Pam$_3$CSK$_4$ concentration is based on previously known mediation of platelet–neutrophil aggregates[8]; low concentration of thrombin was used in order to activate platelets without making them form a thrombus. Platelets do not tolerate DMSO at concentrations higher than 0.05%, thus, loxoribine was dissolved in 700 μL DMSO$_4$ and 770 μL water. Cells were treated with 1 mM loxoribine; complement C3 and compstatin were dissolved in HEPES-modified Tyrode's buffer.

**In vitro experimental conditions**. All in vitro experiments in this study were performed at 37 °C and constant rotation of 1000 rpm in a PAP8 Platelet Aggregation Profiler (Bio/Data Corp, PA, USA) aggregometer for the indicated time.

**Mouse models**. All procedures were approved by the University of Massachusetts Institutional Animal Care and Use Committee (protocol # 2324) and conducted accordingly. TLR7 KO mice were originally obtained from S. Akira and then backcrossed to C57BL/6J (WT) for at least 10 generations[32]. C57BL/6J mice were purchased from the Jackson Laboratory (ME, USA, cat# 000664). These studies used sex- and (12–16 weeks) age-matched mice. In the antiplatelet experiment, age-matched cages of WT mice (12 weeks) were randomly assigned to each group. No blinding was used in the analysis of the experimental effect in the models.

**TLR7 specificity**. The TLR7 agonist loxoribine was dissolved in DMSO$_4$, diluted in phosphate-buffered saline, and then injected intraperitoneally at 2.5 μg/g of body weight[6]. Saline control contained an equivalent amount of DMSO$_4$. A second set of mice was inoculated intranasally with 40,000 pfu of influenza A virus (PR8 strain, Charles River, Wilmington, MA, USA) in 30 μL of PBS. Mice were euthanized by carbon dioxide (CO$_2$) asphyxiation at 24 h post-treatment and blood was collected by cardiac puncture. An aliquot of blood was fixed with BD FACS Lysing Solution (BD Biosciences, NJ, USA, cat# 349202) for microscopy. Platelets and plasma were isolated from the rest of the blood sample as described below.

**Platelet depletion mouse model**. Platelets were depleted from C57BL/6J mice as previously described[6]. Briefly, mice were injected intraperitoneally with anti-platelet glycoprotein Ib beta chain (GPIb, CD42b, cat #R300) or anti-immunoglobulin-G (IgG, cat# C301) antibodies (Emfret Analytics, Germany) at 4 μg of antibody/gram of mouse. At 24 h post ablation, mice were infected with influenza A as described in the previous section. Mice were euthanized by CO$_2$ asphyxiation at 24 h post-infection, 5–6 days post-infection and 12 days post-infection, and blood was collected by cardiac puncture.

**Blood collection**. Human blood was drawn by phlebotomy in ACD Solution A (yellow top) tubes (Fisher Scientific, USA, cat# 02-684-26) or in BD Vacutainer

CPT tubes (Fisher Scientific, cat# 02-685-125). Human blood was drawn from healthy donors who were not on any medication for at least 7 days prior to the draw[6]. Blood from influenza-infected patients was drawn from adults who presented with influenza-like illness to UMass Memorial Medical Center from 2016 to 2018, tested positive for influenza A or B by rapid antigen and/or by qPCR, and had <7 days illness. All procedures were approved by the University of Massachusetts Institutional Review Board (protocol # H00009277 and 14268-10) and participants signed informed consent whenever required by IRB.

Murine blood was drawn via cardiac puncture and collected in Citrate-Phosphate-Dextrose (CPD) buffer (16 mM anhydrous citric acid, 102 mM trisodium citrate, 18.5 mM $NaH_2PO_4$, 142 mM D-glucose, pH 7.4). For all murine blood collections, 500 µL of blood was drawn into 200 µL of CPD.

**Isolation of human blood components.** Human platelets were isolated from venous blood into yellow top ACD Solution A tubes. Platelets were isolated as previously described[6]. Briefly, citrated blood was centrifuged at $150 \times g$ for 17 min. Seventy-five percent of the top layer was removed and diluted with platelet wash buffer (10 mM sodium citrate, 150 mM sodium chloride, 1 mM EDTA, 1% (w/v) Dextrose, supplemented with 100 ng/mL PGE1), and centrifuged at $460 \times g$ for 17 min at room temperature. The resulting pellet was resuspended in pre-warmed HEPES-modified Tyrode's buffer (140 mM NaCl, 6.1 mM KCl, 2.4 mM $MgSO_4 \cdot 7H_2O$, 1.7 mM $Na_2HPO_4$, 5.8 mM Sodium HEPES, supplemented with 0.35% BSA and 0.1% Dextrose). Isolated platelets from influenza-infected patients were lysed in 700 µL QIAzol® (included in miRNeasy Mini Kits—see manufacturer information below). Platelet number was determined by means of a blood cell analyzer (Beckman Coulter Ac.T8, CA, USA). Contamination of the platelet preparation was found to be <1 in 50,000.

The human neutrophil/granulocyte population was isolated from freshly drawn venous blood in BD Vacutainer CPT tubes. Blood was centrifuged for 30 min at $1800 \times g$. The top layer was removed and the separation media layer (composed of thixotropic polyester gel and a FICOLL™ Hypaque™ solution) was removed by using a 1 mL pipet tip. The bottom 3 mL fraction was transferred to a new tube and washed once with 10 mL 1× PBS (no $Ca^{2+}$ or $Mg^{2+}$) for 10 min at $350 \times g$, low brake. The supernatant was removed post centrifugation and the red blood cells (RBCs) from the granulocyte fraction were lysed with 5 mL of RBC Lysis Buffer by Roche Diagnostics (Fisher Scientific, cat# 50-100-3296) that had been pre-warmed to 37 °C and gently mixed in a 1:2 proportion, then warmed in a 37 °C water bath for 8 min followed by centrifugation at $250 \times g$ for 5 min, low brake. The lysed solution was removed and the pellet was washed in 10 mL of 1× PBS at $250 \times g$ for 5 min, low brake. The lysis step was repeated with an additional 5 mL of RBC Lysis Buffer to ensure effective RBC lysis. The bottom pellet was again washed in 10 mL of PBS at $250 \times g$ for 5 min, the supernatant was removed, and the white pellet resuspended in HEPES-modified Tyrode's Buffer. Neutrophils were 98–100% platelet-free unless platelets were attached to neutrophils; the neutrophil population had no mononuclear cell presence (Supplementary Fig. 8).

Of note, isolating a platelet-free population of neutrophils by standard FICOLL™ gradient protocols[9] presented a major challenge for our experiments since it was necessary to obtain platelet-free neutrophils (Supplementary Fig. 8). With modifications (procedure is described under human neutrophil/granulocyte) we were able to establish an isolation method free of platelets ($0 \times 10^3/\mu L$, as measured by the blood cell analyzer). Confocal microscopy showed that in some cases platelets were attached to neutrophils (heterotypic aggregates) and could not be removed. We were not able to find a negative depletion neutrophil prep that eliminates platelets (containing either CD41 or CD42 antibodies) or neutrophils with attached platelets. Additionally, neutrophils were treated gently without vortexing to minimize bubbles. This was necessary to ensure that neutrophils were not activated and/or primed by released platelet content during isolation steps. Standardization of methods of platelet-free neutrophil isolation and treatment may be necessary to delineate the role of platelets and/or neutrophils during the initial steps of physiological infection.

**Human (influenza patients) and mouse plasma isolation.** Human blood (1 mL) or mouse blood (600 µL) in CPD was centrifuged at $500 \times g$ for 10 min. The supernatant was removed, centrifuged at $2000 \times g$ for 10 min and the plasma was immediately frozen on dry ice.

**Mouse platelet isolation for influenza RNA qPCR.** Mouse blood was collected in CPD buffer as described above. Blood was centrifuged at $500 \times g$ for 10 min. The supernatant was removed for further centrifugation for plasma collection as described above. The remaining blood pellet was gently transferred to a Falcon® 5 mL polystyrene round-bottom tube (Corning, NY, USA, cat# 352058) that contained 4 mL of CPD buffer and centrifuged at $300 \times g$ for 5 min, no brake. The supernatant (leaving a small volume above blood pellet) was gently transferred to a second Falcon® 5 mL tube, the tube was filled to the top with CPD, and centrifuged at $3500 \times g$ for 7 min, no brake. The supernatant was removed, any remaining RBCs were manually removed from the platelet pellet with a pipette and the platelets were resuspended in QIAzol® lysis reagent.

**Influenza A strain (WSN/33) sucrose gradient purification.** Infectious influenza A viral strain WSN/33, was purified and concentrated as previously described[63]. The final viral pellet was resuspended in chilled DPBS, aliquoted and stored at −80 °C.

**Confocal microscopy and antibodies.** Whole blood: Whole blood from in vitro incubated experiments, influenza-infected patients, or mice was lysed and fixed with 1× BD FACS™ lysing solution (BD Biosciences, cat #349202) for 10 min. Lysed and fixed blood from influenza-infected patients was stored at 4 °C at this step until processing. To ensure possible effects of fixation and labeling on blood cells, we processed the blood from infected and uninfected controls identically; in all cases antibody labeling was always done after fixation to avoid unwanted interactions between the antibody and the platelets. Samples were centrifuged at $1100 \times g$ for 7 min, washed once with 1 mL of HEPES-modified Tyrode's buffer under the same conditions, resuspended in 100 µL of HEPES-modified Tyrode's buffer supplemented with 2% FBS and blocked for 1 h at room temperature, then antibodies were added for an additional hour. At the end of incubation, samples were washed with 1 mL of 1× PBS and mounted on slides. Treatment of unfixed blood or isolated cells in the presence of antibodies led to the engulfment of platelets by neutrophils similarly to the in vivo antibody depletion data in Fig. 9a (yellow stain of neutrophils). To eliminate the effect of opsonization-mediated interactions, we performed all microscopy staining after fixation post-treatment.

Intracellular staining of isolated or influenza-incubated platelets: Platelets in solution (at $2 \times 10^5$ platelets/µL in 100 µL) were brought to 1 mL with HEPES-modified Tyrode's buffer at constant rotation of 1000 rpm in an aggregometer (PAP-8). 333 µL of 16% paraformaldehyde was added and platelets were fixed for 10 min (at 1000 rpm). Tubes were removed from the aggregometer, placed at room temperature on a stir plate at 1000 rpm and 1% Saponin was added to the mixture for 7 min. Samples were washed with 1 mL washing buffer (0.4% BSA; 0.1% Saponin in PBS) and centrifuged at $1800 \times g$ for 5 min. Samples were then resuspended in staining buffer (1% BSA; 0.1% Saponin in HEPES-modified Tyrode's buffer) that contained the antibodies of interest. Samples were incubated for 1 h at 4 °C, washed with 1 mL of 1× PBS at $1800 \times g$ for 5 min and mounted on slides.

The following antibodies (in 100 µL of staining solution) were used throughout this study: anti-human: 10 µL CD41-FITC or 8 µL CD41-APC (clone HIP8, eBioscience, CA, USA, cat# 11-0419 and cat# 17-0419), 5 µL CD66b-APC (clone G10F5, eBioscience, cat# 17-0666), 5 µL MPO-FITC (clone MPO455-8E6, eBioscience, cat# 11-1299), 2 µL Histone H4-AF647 (clone 31830, Abcam, MA, USA, cat# ab197515, also recognizes mouse), 2 µL Histone H3 (Abcam, cat# ab1791), followed by FITC-conjugated Goat Anti-Rabbit IgG H&L secondary antibody (Abcam, cat# ab6717); 2 µL TLR7-APC (clone 4G6, Novus Biologicals, cat# NBP2-25274APC), 5 µL LAMP-1/CD107a -DyLight 405 (clone 5E7, Novus Biologicals, cat# NBP2-52721V), 5 µL CD63-BV421 (clone H5C6, Biolegend, cat #353029); anti-mouse: 10 µL CD41-FITC (clone MWReg30, eBioscience, cat# 11-0411), 5 µL Ly6G-APC (clone RB6-8C5, eBioscience, cat# 17-5931); and 2 µL Influenza A-NP-FITC (Abcam, cat# ab20921), 2 µL Influenza B-NP-FITC (Invitrogen, cat# MA1-7306). Mounted slides were resolved by fluorescent microscopy using a Spinning Disk Confocal Nikon TE2000E2 inverted microscope or Scanning Disk Nikon A1 confocal microscope.

**Transmission electron microscopy.** Platelets incubated with influenza in vitro: Isolated platelets were incubated with influenza for various amounts of time in 100 µL of HEPES-modified Tyrode's buffer, at constant rotation in an aggregometer (PAP-8). After incubation the solution was brought to 500 µL with the same buffer and then fixed for 10 min at 37 °C and constant rotation with 500 µL of Karnovsky's fixative (a mixture of 2.0% paraformaldehyde and 2.5% glutaraldehyde in 0.1 M Sorensen's phosphate buffer, pH 7.4), before initial centrifugation and were immediately processed.

Platelets from influenza-infected patients and control donors: Isolated platelets (as described in Human (influenza patients) and mouse plasma isolation) were resuspended in 200 µL CPD and then fixed and stored with Karnovsky's fixative, for not more than a month, before processing.

In either case, samples were further processed and resolved by a Philips CM10 electron microscope (Eindhoven, Netherlands) as previously reported[6].

**Mixing experiments using platelets and or neutrophils.** All mixing experiments were done using $2 \times 10^5$ platelets/µL and $0.04 \times 10^5$ neutrophils/µL, or a ratio of 1 neutrophil:50 platelets. All mixing experiments were conducted in a platelet appropriate environment using HEPES-modified Tyrode's buffer[6] supplemented with 3 mM fibrinogen (Enzyme Research Labs, IN, USA, cat # FIB3) and 1 mM $Ca^{2+}/Mg^{2+}$ in a final volume of 225 µL, and were carried out at 37 °C and constant rotation of 1000 rpm in an aggregometer (PAP-8).

To test the contribution of platelets to the release of neutrophil-DNA, isolated platelets or isolated neutrophils were pretreated with TLR agonist for 15 min and added to the respective untreated cell-population. Cells were co-incubated for 30 min then fixed with eBioscience IC fixation buffer (ThermoFisher Scientific, MA, USA, cat# 00-8222). To test the role of platelets and neutrophils in cytokine release, platelets, neutrophils, and a mixture of both cell types were treated with TLR agonist for 30 min. Upon completion, cells were centrifuged (7 min, $1000 \times g$, room

temperature), and supernatants frozen for analysis by ELISA. To test whether C3 can mediate NETosis, neutrophils were treated with C3 (30 ng/mL, resuspended in HEPES-modified Tyrode's buffer), GM-CSF (25 ng/mL, resuspended in HEPES-modified Tyrode's buffer) or both for 30 min. Platelets were centrifuged and supernatant frozen. For confocal microscopy, neutrophils and the mixture of both cell types were fixed with IC fixation buffer.

To establish if the platelet-TLR7-C3 axis mediates the release of DNA from neutrophils, platelets, neutrophils, and a mixture of both were pretreated with 0.088 mg/mL of compstatin (C3 inhibitor) for 10 min and then treated with TLR7 agonist for 30 min. Platelets were centrifuged at $1000 \times g$ for 7 min and the supernatant frozen. Neutrophils and the mixture of both cell types were fixed with IC fixation buffer.

To examine whether influenza leads to TLR7 mediated release of C3 from human platelets, isolated platelets were pre-treated in the presence or absence of TLR7-antagonist IRS661 (synthesized by Eurofins, resuspended in TE buffer, final concentration 2.8 µM) and then incubated with infectious virions of influenza A (WSN/33) for 30 min (at a ratio of 1 pfu:10 platelets; 1 pfu:100 platelets; or 1 pfu:1000 platelets). Aliquots of the treated platelets were centrifuged at $1100 \times g$ for 5 min and the supernatant was aliquoted and frozen.

Of note, throughout this study we have used an in vitro incubation time of 30 min to assess platelet responsiveness and neutrophil-DNA release. Our previous studies[6] show that platelets and neutrophils can form heterotypic aggregates after 15 min of stimulation with TLR7 agonist. Here we wanted to assess how this interaction is related to neutrophils at an early time point of infection.

**ELISAs**. 50 µL of supernatant collected following cell mixing treatment was assayed for GM-CSF (Ebioscience, cat# 88-8837 or Abcam, cat# ab174448). 1 µL of supernatant collected after cell mixing treatment was assayed for C3 (Abcam, cat# 108823); human plasma diluted 800× was assayed for C3 (Abcam, cat# 108822); murine plasma was diluted 50,000× (Abcam, cat# 157711); human plasma diluted 100× was assayed for Neutrophil Elastase (Abcam, cat# ab119553), MPO (Abcam, cat# ab119605), and histone nucleosome core (NovusBio cat# KA1091). 10 µL of plasma isolated after ex vivo treatment and 5 µL (20 µL for platelets only) of supernatant collected after cell mixing treatment were assayed for MPO (Abcam, cat# ab119605); 50 µL of plasma after ex vivo treatment was assayed for IFN-alpha (Abcam, cat# ab213479); 40 µL of plasma after ex vivo treatment was assayed for IL-8 (Ebioscience cat# 88-8086-22); and 40 µL of plasma after ex vivo treatment was assayed for citrullinated H3 (Cayman Chemical, MI, USA, cat# 501440); human plasma of influenza-infected patients and platelets supernatants were diluted 25×, and assayed for CCL5 (Abcam, cat# 174446). All procedures were performed according to the manufacturers' instructions.

**RNA isolation from human and mouse platelets**. Frozen RNA in QIAzol® was thawed at room temperature for 60 min at 2000 rpm on an Eppendorf MixMate plate shaker (Eppendorf, Germany, cat# 022674200). Total RNA was isolated from human and mouse platelets using the miRNeasy® Mini Kit (Qiagen, Germany, cat# 217004) following the manufacturer's instructions with on-column DNA digestion using the RNase-Free DNase Set (Qiagen, cat# 79254) and eluted in 30 µL of RNase-free water. RNA concentration was determined via a NanoDrop Spectrophotometer (ThermoFisher Scientific, MA, USA, model# ND-1000), or via Fragment Analysis by the Molecular Biology Core Lab at the University of Massachusetts Medical School.

**Detection of influenza RNA and TLR7 mRNA by RT-qPCR**. Platelet complementary DNA (cDNA) was synthesized using the High Capacity cDNA RT Kit (Applied Biosystems, CA, USA, cat# 4368813) in a 10 µL reaction volume (10× Reverse Transcription Buffer [1 µL], 25× dNTPs [0.4 µL], 10× Random Primers [1 µL], and Multiscribe Reverse Transcriptase, 50 U/L [0.5 µL]) with a maximum of 7.1 µL of RNA for humans or a 20 µL reaction volume (10× Reverse Transcription Buffer [2 µL], 25× dNTPs [0.8 µL], 10× Random Primers [2 µL], and Multiscribe Reverse Transcriptase, 50 U/L [1 µL]) with a maximum of 14.2 µL of RNA for mice. RNA was either normalized to the sample with the lowest RNA concentration or a specific volume of RNA was used. cDNA synthesis was performed on a thermal cycler (Applied Biosystems, Veriti 9903 or ProFlex) under the following conditions: 25 °C for 10 min, 37 °C for 2 h, 4 °C hold.

cDNA was preamplified using TaqMan™ PreAmp Master Mix (Applied Biosystems, cat# 4391128) in a 5 µL reaction volume (Master Mix [2.5 µL], 0.2× assay pool [1.25 µL]) with 1.25 µL of cDNA for humans or in a 40 µL reaction volume (Master Mix [20 µL], 0.2× assay pool [10 µL]) with 10 µL of cDNA for mice. A standard curve was also preamplified using influenza DNA plasmid (PR8) for the human samples. The plasmid, of known base pair size and DNA concentration, was serially diluted in RNase-free water and preamplified. Preamplification was performed on a thermal cycler as listed above, under the following conditions: 95 °C for 10 min, 14 cycles of 95 °C for 15 s and 60 °C for 4 min, 4 °C hold. For human samples, the final preamplification product was diluted 1:9 in DNA Suspension Buffer (Teknova, CA, USA, cat# T0223), prior to qPCR. For mouse samples, the preamplification product was not diluted.

Influenza gene expression in humans and mice was quantified by RT-qPCR (Applied Biosystems, 7900HT Fast or QuantStudio3 Fast Real-Time PCR systems)

using TaqMan™ Gene Expression Master Mix (Applied Biosystems, cat# 4369016) and TaqMan™ Gene Expression Assays (see table below for TaqMan assays used) in a 40 µL reaction volume on the 7900HT system (Master Mix [20.0 µL], TaqMan Assay [2 µL], diluted pre-amplification product [18 µL]) or a 30 µL reaction volume on the QuantStudio3 system (Master Mix [15.0 µL], TaqMan Assay [1.5 µL], diluted pre-amplification product [13.5 µL]), under the following conditions: 50 °C for 2 min, 95 °C for 10 min, and 40 cycles of 95 °C for 15 min and 60 °C for 1 min.

TLR gene expression in humans was quantified as above but in a 10 µL reaction volume (Master Mix [5.0 µL], TaqMan Assay [0.5 µL], diluted pre-amplification product [4.5 µL]). TaqMan gene expression assays used are listed in Supplementary Table 3.

**Calculation of influenza virus copy number**. Copy number was calculated based on previously published methods[64,65], and using linearized influenza plasmid DNA (PR8) used to generate a standard curve to calculate viral copy number by qPCR.

**Detection of free DNA**. Detection of DNA in 2 µL of citrated plasma samples was performed using the Quant-iT™ PicoGreen® dsDNA reagent kit and Lambda DNA standard (Invitrogen, CA, USA, cat# P7589) according to the manufacturer's instructions. Briefly, 98 µL of TE buffer were aliquoted in 96-well plate, 2 µL of plasma were added followed by 100 µL of PicoGreen® dsDNA reagent. Fluorescence was measured on a POLARstar Omega microplate reader (BMG Labtech, Germany) at an excitation wavelength of 480 nm and an emission wavelength of 520 nm. Plasma DNA concentration was determined by plotting fluorescence on the Lambda DNA standard curve.

**Statistical analysis**. All FHS data analysis (Supplementary Data) was done using linear mixed effects models (the "LMEKIN" function of Kinship Package in R) with an additive genetic model to test associations of inverse-rank normalized protein levels with SNPs. To account for multiple testing, we applied Bonferroni correction to attain acceptable type I error rates. All other data were analyzed using GraphPad Prism 5 or 7 and details can be found in the legend of each figure.

**Reporting summary**. Further information on experimental design is available in the Nature Research Reporting Summary linked to this article.

## Data availability

The source data underlying Figs. 1e–g, i, 5a–d, 6a–c, e, 7b, c, e, 8a, b, 9b–d, 10b–d and Supplementary Figs. 3, 9a-b, 15, 16b and 18a are provided as a Source Data file. The ELISA data for the human patients and the transmission electron microscopy and fluorescent images included in the supplementary data have been deposited to Dryad database, https://doi.org/10.5061/dryad.786b9q0. The authors declare that the data supporting the findings of this study are available within the article and its Supplementary Information files, or are available from the authors upon request.

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

## Acknowledgements

The authors thank Prof. Sanjay Ram, MD (UMass Medical School) for his helpful advice and discussion on complement C3. The authors thank K. Longtine and J. Longtine for patient blood collection and Michael King for propagation, titration, and sucrose-gradient purification of the WSN/33 flu strain. This work was supported by AHA grant 16SDG30450001 (to M.K.) and NIH grants N01-HC 25195, U01HL126495, UH3TR000921-04 and a supplement to UH3TR000921-04 provided by the NIH Common Fund (to J.E.F.).

## Author contributions

M.K. and J.E.F. designed, interpreted the results, and wrote this article. M.K. conducted all of the experiments; H.A.C. provided assistance with the co-mixing experiments and with G.M. and L.C. conducted qPCR; O.V. performed the TEM sample processing and imaging

in the UMass Core Electron Microscopy Facility. R.W.F. and J.P.W. assisted with logistics for influenza patient blood collection; E.A.K.-J. provided all influenza stocks and together with assistance of C.J.P. performed the initial influenza infections in mice; J.P.W. provided the PR8 plasmid and the standard curve calculations; C.Y. and D.L. provided the association assessment of inverse-rank normalized protein levels with SNPs; J.R. provided samples. All authors assisted with the editing and revision of the manuscript.

## Additional information

**Competing interests:** The authors declare no competing interests.

