## [Peer Review File · Nature Communications]

Reviewers' Comments:

Reviewer #1:

Remarks to the Author:

Journal Name: Nature Communication

Manuscript Number: 167758_0_supp_3016666_p8fyyf_convrt

Title of the Manuscript: The Role of Platelets in Mediating a Response to Human Influenza Infection

A manuscript entitled "The Role of Platelets in Mediating a Response to Human Influenza Infection" has been submitted for consideration to Nature Communication.

The authors try to determine if platelet processes are activated during influenza infection. The authors concluded that the initial intrinsic defense against influenza is mediated by platelet-neutrophil cross communication that tightly regulates host immune and complement responses but can also lead to thrombotic vascular occlusion.

This is a very interesting and remarkable paper; the manuscript is well-written and should be of great interest to "Platelet-Related inflammation in an infectious context" scientific community. Platelets are non-nucleated cells that play central roles in the processes of hemostasis, innate immunity, and inflammation; however, several reports show that these distinct functions are more closely linked than initially thought. This study examines the role of platelets in anti-infection immunity via their TLRs.

- The major Comments concerns

1. Polymorphisms in TLR7 have been linked to susceptibility to HCV infection in the Taiwanese population (Wang et al 2011), also it would be necessary to study if different TLR7 polymorphism influence the defense against influenza mediated by platelet-neutrophil cross communication.

2. Moreover, Koupenova et al report an interesting study to demonstrate that platelets express all TLR transcripts using a large community-based observational cohort. Koupenova et al found that platelets express all TLR transcripts at variable levels and, surprisingly, women had higher levels of all known TLRs (1-10). It would be necessary to study if association of sex on TLR7 expression modulate the defense against influenza mediated by platelet-neutrophil cross communication.

3. "Heterotypic aggregates between platelets and leukocytes are observed during gram positive bacterial infections as well as during infections with single-stranded viruses such as encephalomyocarditis virus (6, 8)." The authors must describe gram negative bacterial and platelet interaction .

4. A second set of mice was inoculated intranasally with 40,000 plaque-forming units (pfu) of influenza A virus (PR8 strain, Charles River, Wilmington, MA, USA) in 30 μ L of PBS. The authors must insert data, concerning an inoculation of another pfu of influenza A virus (clinical strain).

5. The authors must discuss some of the hypotheses that may explain how platelets may control a Response to Human Influenza Infection comparing to other infection and non-infectious stress.

- The minor Comments concerns

1. "Thrombin and Pam3CSK4 were dissolved in water and blood or isolated cells were treated with 10 μ g/ μ L of Pam3CSK4 or 0.05 U/mL of thrombin". The authors must insert data as supplemental data,

concerning the choice of 10 µg/µL of Pam3CSK4 or 0.05 U/mL of thrombin.

2. "Neutrophils were 98-100% platelet-free unless platelets were attached to neutrophils". The authors should in this study define, in the discussion section, the potential inflammatory role of (hypothetical) blood platelets attached to neutrophils.
3. "To test the role of platelets and neutrophils in cytokine release, platelets, neutrophils and a mixture of both cell types were treated with TLR agonist for 30 min. Upon completion, cells were centrifuged (7 min, 1000 x g, room temperature), and supernatants frozen for analysis by ELISA." The authors must insert data as supplemental data, concerning the choice of the activation length '30 min'
4. "To examine whether influenza leads to C3 release from human platelets, isolated human platelets were treated with infectious virions of influenza A (WSN/33) for 30 min (at a ratio of 4 pfu to 1000 platelets)." The authors must insert data as supplemental data, concerning the choice of ratio 4 pfu to 1000 platelets.
5. "ELISA": The authors must detail the choice of IFN alpha and IL-8 assays.
6. "TaqMan Gene Expression Assays Used": The authors must indicate several technical reference for each paragraph.
7. The authors must detail the choice of Statistical test.
8. The authors must indicate the agreement number concerning the ethics of animal and human research.
9. Finally, future direction paragraph should be inserted because future directions may also include interesting next steps in the research.

Reviewer #2:

Remarks to the Author:

Koupenova M et al showed that influenza virus infection activates TLR7-expressing platelet, leading to C3 release, promoting NETs formation and neutrophil-DNA-platelet aggregates. In particular, the authors observed by confocal microscopy, the interaction between platelets and released neutrophil DNA in the blood of patients infected with influenza virus. Using transmission electron microscopy, the authors showed that influenza particles are present within platelets. Interestingly, when platelets from healthy donors are incubated with influenza virus (WSN/33) or activated with a TLR7 agonist (Ixo), C3 is released by platelets (ELISA assay) that express TLR7, leading to neutrophil-DNA release. Finally, the authors used a very elegant model showing that in vivo stimulation of WT mice with a TLR7 agonist or infection of these mice with influenza virus, induce NETs release in the blood. This effect was not seen using TLR7-KO mice showing that the mechanism occurs through TLR7 in vivo, as well. In addition, in vivo platelet depletion with the anti-CD42 antibody abrogates this effect, showing the role of platelets in these settings.

In overall, this manuscript is of high interest, novel and is very easy to read. However, below are some comments for improvements:

- 1) Since the focus of the paper is to demonstrate that platelets are infected by influenza virus, which can then activate TLR7, one required experiment is to clearly demonstrate the incorporation of

influenza virus into platelets (in vitro) and define the localization of TLR7 (granules?). For this, authors could incubate platelets with influenza virus and at different time points label the virus within the platelets and TLR7.

2) Line 136: we are not convinced by the qRT-PCR that was observed in 50% of patients in 2016. Indeed, qRT-PCR could easily amplify the influenza virus genes from leucocytes contaminate, existing in the samples.

3) In figure 1C and D, the labeling of neutrophils is required. Indeed, NETs are known to be released by many cell types such as monocytes (Granger et al. 2017).

4) Figure 1E is not convincing. We agree that the second panel showed the presence of NETs in the blood of infected patients and colocalization with platelets. However the two other panels do not demonstrate these interactions. In the third panel, platelets are not present at all (no green colabeling) and in the fourth panel, the round circle cannot be representative of NETs and the red histones do not colocalize with platelets and DNA.

5) Maybe a higher resolution is required for Figure 2B. As shown here, we cannot really distinguish whether the authors point the presence of influenza virus into platelets or simply some vesicles. Accordingly, it is necessary to add a positive control: platelets isolated from a donor and infected in vitro.

6) Figure 3A: it is necessary to show a figure with all the controls: (i) uninfected platelets that are devoid of TLR7, (ii) infected platelets that are devoid of TLR7, (iii) uninfected platelets that express TLR7 and (iv) infected platelets that express TLR7.

7) Figure 6 A: Results for TLR7-KO mice need to be shown at 4h and at 24 hours, WT + loxo and TLR7-KO + saline need to be shown.

8) it is absolutely required to perform an experiment with infected platelets in presence of an inhibitor of TLR7 and observe an abrogation of C3 release and NETs formation.

9) Figure 6C: why the authors did not performed an anti-neutrophil-Ly6G instead of an anti-H3 (to be homogene with figure 6A)?

In addition, here are minor comments:

- please note in Figure 1, how DNA was labelled
- In line 220, the authors noted no "significant" changes, while in Figure 5A, some differences can be observed.
- please add in the material and methods the clones used for anti-CD41 APC and anti-H3 antibodies
- Figure 7B: we do not see the importance of comparing the significativity between igG + flu and aPlt + sal. Just add "non significant" between the depleted mice uninfected and infected.
- line 267: modify the sentence. The virus activates TLR and release of C3, not the platelet
- The authors over interpreted the results. While the concept is very interesting, the authors did not show any experiment regarding the elevated risk of myocardial infraction. This should only be briefly discussed.

Reviewer #3:

Remarks to the Author:

Koupenova et al. conducted a series of experiments using human blood samples and mouse models to identify the relevance of NET formation in the context of platelets, C3, influenza viral infection and myocardial infarction. The authors show that flu patients have DNA-platelet aggregates in the circulation, and platelet TLR7 activation is important for DNA release from neutrophils in vitro. The mouse model data suggest that TLR7 is needed for viral clearance and perhaps important for extracellular trap release.

Following points need to be addressed to strengthen the conclusions made in the manuscript.

Major

1. The intro cites references to highlights the importance of monocytes in MI, but the study focusses on neutrophils.
2. No data is provided to support the hypothesis and the relevance of flu, NET formation and myocardial infarction (MI). Therefore, the hypothesis requires a through validation/revision.
3. The paper talks about initial flu infection. Sufficient data are not provided to support the role of initial stage flu infection to DNA release and MI.
4. Several technical issues need to be addressed – no clear proof for NETs, image analyses, quantifying NETs and statistical analyses.
5. Often an image of a magnified cell is shown to support the data, without quantifying the data set. This is insufficient to make a claim with scientific rigor.
6. Several statistical analyses are inappropriate. The t-test cannot be used for comparing more than one set of values. Also, it is incorrect to compare a constant value (control value=1) with other values by t-test. Furthermore, the values that are not significantly different are stated as different (Fig. 7C). Therefore, the conclusions made in several experiments are incorrect.
7. The claim that the "DNA seen in the circulation is NET" is incorrect because no data is provided to support that the DNA shown in the images belongs to neutrophils (H4 is not a neutrophil marker). The authors also state that the DNA detected in the blood does not contain NET markers. Also, MPO is only "sometimes" present on DNA, suggesting that the process studied here is not NETosis.
8. The authors suggest that they have identified a special form of NETosis, but without providing convincing data sets to support their claim.
9. C3 is also present in neutrophils. The authors only considered C3 of platelets. Therefore, the interpretation of this data set is also inappropriate.
10. TEM images with no immunostaining and quantitative data are not reliable.

Minor

11. The introduction is very long. Many points highlighted here belongs to discussion.

We thank the reviewers for their insightful comments and taking the time to critically evaluate our manuscript. We have addressed their concerns, and this has improved our findings and overall manuscript.

Reviewers' comments:

Reviewer #1 (Remarks to the Author):

Journal Name: Nature Communication

Manuscript Number: 167758_0_supp_3016666_p8fyf_convrt

Title of the Manuscript: The Role of Platelets in Mediating a Response to Human Influenza Infection

A manuscript entitled "The Role of Platelets in Mediating a Response to Human Influenza Infection" has been submitted for consideration to Nature Communication.

The authors try to determine if platelet processes are activated during influenza infection. The authors concluded that the initial intrinsic defense against influenza is mediated by platelet-neutrophil cross communication that tightly regulates host immune and complement responses but can also lead to thrombotic vascular occlusion.

This is a very interesting and remarkable paper; the manuscript is well-written and should be of great interest to "Platelet-Related inflammation in an infectious context" scientific community. Platelets are non-nucleated cells that play central roles in the processes of hemostasis, innate immunity, and inflammation; however, several reports show that these distinct functions are more closely linked than initially thought. This study examines the role of platelets in anti-infection immunity via their TLRs.

We thank the reviewer for the supportive words. We now provide further evidence that enhances the previously outlined conclusions.

• *The major Comments concerns*

1. Polymorphisms in TLR7 have been linked to susceptibility to HCV infection in the Taiwanese population (Wang et al 2011), also it would be necessary to study if different TLR7 polymorphism influence the defense against influenza mediated by platelet-neutrophil cross communication.

Our current data is not suitable to analyze a polymorphism and its association with disease (due to size). However, we analyzed the above mentioned polymorphism using the FHS (Offspring Cohort, Visit 8). C3 has not been measured in this population but there are measurements of C2 in the plasma of this cohort. Thus, we analyzed the mentioned (in the paper, Wang et al 2011) polymorphisms with the pQTL of C2 in blood cells in FHS. C2 is upstream of the C3 component of the complement system necessary for the processing of C3. In this population, there were no pQTLs of these two SNPs for

C2 at $P < 1e-6$ in this cohort. Future studies, beyond the scope of ours, are necessary to address the reviewer's question. These observations are included in Supplemental Data.

2. Moreover, Koupenova et al report an interesting study to demonstrate that platelets express all TLR transcripts using a large community-based observational cohort. Koupenova et al found that platelets express all TLR transcripts at variable levels and, surprisingly, women had higher levels of all known TLRs (1-10). It would be necessary to study if association of sex on TLR7 expression modulate the defense against influenza mediated by platelet-neutrophil cross communication.

This is an interesting suggestion. In this limited population, we do not see differences by sex when it comes to C3 release in the influenza-infected patients (data not shown). We agree that it would be fascinating to study sex differences in regards to infection and hope to do so in future (larger) studies.

3. "Heterotypic aggregates between platelets and leukocytes are observed during gram positive bacterial infections as well as during infections with single-stranded viruses such as encephalomyocarditis virus (6, 8)." The authors must describe gram negative bacterial and platelet interaction.

We have corrected the sentence to include gram-negative bacteria mediated interactions:

"Heterotypic aggregates between platelets and neutrophils are observed during gram-positive bacterial infections, with gram-negative bacterial components as well as during infections with single-stranded viruses such as encephalomyocarditis virus (1-3)."

4. A second set of mice was inoculated intranasally with 40,000 plaque-forming units (pfu) of influenza A virus (PR8 strain, Charles River, Wilmington, MA, USA) in 30 μ L of PBS. The authors must insert data, concerning an inoculation of another pfu of influenza A virus (clinical strain).

Mice do not naturally exhibit the symptoms and inflammation as found in humans. Thus, we can only use a mouse-adapted strain (PR8) at a high pfu to induce a partially biosimilar response to humans. Even following a high influenza inoculation, mice recover from infection. Using clinical strains in mice would not lead to observable infection.

5. The authors must discuss some of the hypotheses that may explain how platelets may control a Response to Human Influenza Infection comparing to other infection and non-infectious stress.

We have included the following paragraph in the discussion on pg. 18-19:

Interestingly, during sepsis, C3 proteolysis is predictive of the severity of infection and sepsis can lead to MI (41, 42). Perhaps circulating neutrophil-DNA-platelet aggregates, although necessary for viral (or bacterial) removal and adaptive immunity during infection, may become pathological throughout the course of infection with increased vascular damage and possible MI.

- *The minor Comments concerns*

1. *“Thrombin and Pam3CSK4 were dissolved in water and blood or isolated cells were treated with 10 µg/µL of Pam3CSK4 or 0.05 U/mL of thrombin”. The authors must insert data as supplemental data, concerning the choice of 10 µg/µL of Pam3CSK4 or 0.05 U/mL of thrombin.*

We thank the reviewer for pointing this out. There was a misstatement in that the final concentration of Pam3CSK4 is 10 µg/mL and this concentration was selected because it is known to lead to platelet-neutrophil aggregates (1). A low concentration of thrombin was used in order to not overwhelm platelets to form a clot and thereby release their entire contents. This information is included in the Methods section under *“Pharmacological compounds” pg.23.*

2. *“Neutrophils were 98-100% platelet-free unless platelets were attached to neutrophils”. The authors should in this study define, in the discussion section, the potential inflammatory role of (hypothetical) blood platelets attached to neutrophils.*

Although it would be interesting to define the hypothetical role of platelets attached to neutrophils, we are not comfortable with commenting on this observation as no data or studies address exactly what these platelets do. We can only speculate that this is part of normal platelet-neutrophil cross communication in blood, but we have no direct evidence. We also have no evidence for a potential inflammatory role of these aggregates at baseline.

3. *“To test the role of platelets and neutrophils in cytokine release, platelets, neutrophils and a mixture of both cell types were treated with TLR agonist for 30 min. Upon completion, cells were centrifuged (7 min, 1000 x g, room temperature), and supernatants frozen for analysis by ELISA.” The authors must insert data as supplemental data, concerning the choice of the activation length ‘30 min’*

Our previous studies (2) show that platelets and neutrophils can form heterotypic aggregates at 15 min post stimulation with TLR7 agonist. At this time point we have occasionally observed neutrophils with released DNA. We used 30 min in order to increase the incubation time to observe NETosis without compromising platelet function. We have inserted this clarification in the Methods section on pg 31 as follows:

*“Of note, throughout this study we have used an *in vitro* incubation time of 30 min to assess platelet responsiveness and neutrophil DNA release. Our previous studies (2) show that platelets and neutrophils can form heterotypic aggregates after 15 min of stimulation with TLR7 agonist. Here we wanted to assess how this interaction is related to neutrophils at an early time point of infection.”*

4. *“To examine whether influenza leads to C3 release from human platelets, isolated human platelets were treated with infectious virions of influenza A (WSN/33) for 30 min (at a ratio of 4 pfu to 1000 platelets).” The authors must insert data as supplemental data, concerning the choice of ratio 4 pfu to 1000 platelets.*

In the first version of the paper, that was the highest amount of pfu that we could use to recapitulate the observed presence of flu in some of the patients. The viral stock had been collected in media from infected cells. To selectively address the role of the virus on platelets, we have re-done the experiments utilizing sucrose purified and concentrated viral stock. We performed the experiment with 3 different pfu's to assess the TLR7-mediated effect. A high dose of the virus (10 platelets to 1 virion) increased C3 release, but was not specific for the TLR7 receptor. A low dose (1000 platelets to 1 virion) did not induce detectable C3 release in the supernatant at 30 min. 100 platelets to 1 virion led to a TLR7-specific effect that we are reporting here. We know that there are other cytosolic viral mechanisms that can be involved at higher viral concentrations and, at lower concentrations, time might be limiting. Mechanistically, however, our experiment supports the hypothesis that C3 is released as a function of TLR7 from platelets. This information is now included on pg. 9.

5. *“ELISA”: The authors must detail the choice of IFN alpha and IL-8 assays.*

Sequencing data analyzing the transcriptome of small and large platelets shows presence of the IL8 transcript in platelets (4). Since IL-8 is a chemokine that is chemotactic for neutrophils, we wanted to evaluate if platelets can release IL-8 as a result of TLR7 stimulation. We have removed IFN alpha from this manuscript as it was used more as a negative control. IL-8 justification is now included on pg. 13.

6. *“TaqMan Gene Expression Assays Used”: The authors must indicate several technical reference for each paragraph.*

We have provided a reference to the study used to calculate copy number. The reference for the primers comes from the World Health Organization and the link to the website information is provided below the table. See pg. 34.

7. *The authors must detail the choice of Statistical test.*

We have included a section titled “Statistical Analysis” on pg. 35

8. *The authors must indicate the agreement number concerning the ethics of animal and human research.*

We have included the requested information on pg.23, 25.

9. *Finally, future direction paragraph should be inserted because future directions may also include interesting next steps in the research.*

We have included the following future studies paragraph in the discussion, pg.22:

“Future studies may focus on understanding the contribution of other infections to platelet activation and neutrophil DNA release and generalization of this mechanism. It will also be beneficial to understand C3 regulation as a function of non-pathogenic inflammation and to evaluate if regulation of C3 by inhibitors such as compstatin may be beneficial to acute cardiothrombotic events.”

Reviewer #2 (Remarks to the Author):

Koupenova M et al showed that influenza virus infection activates TLR7-expressing platelet, leading to C3 release, promoting NETs formation and neutrophil-DNA-platelet aggregates. In particular, the authors observed by confocal microscopy, the interaction between platelets and released neutrophil DNA in the blood of patients infected with influenza virus. Using transmission electron microscopy, the authors showed that influenza particles are present within platelets. Interestingly, when platelets from healthy donors are incubated with influenza virus (WSN/33) or activated with a TLR7 agonist (loxo), C3 is released by platelets (ELISA assay) that express TLR7, leading to neutrophil-DNA release. Finally, the authors used a very elegant model showing that in vivo stimulation of WT mice with a TLR7 agonist or infection of these mice with influenza virus, induce NETs release in the blood. This effect was not seen using TLR7-KO mice showing that the mechanism occurs through TLR7 in vivo, as well. In addition, in vivo platelet depletion with the anti-CD42 antibody abrogates this effect, showing the role of platelets in these settings.

In overall, this manuscript is of high interest, novel and is very easy to read. However, below are some comments for improvements:

1) Since the focus of the paper is to demonstrate that platelets are infected by influenza virus, which can then activate TLR7, one required experiment is to clearly demonstrate the incorporation of influenza virus into platelets (in vitro) and define the localization of TLR7 (granules?). For this, authors could incubate platelets with influenza virus and at different time points label the virus within the platelets and TLR7.

We thank the reviewer for raising this important point. We have conducted two sets of experiments to address the reviewer's concern.

First, we now provide transmission electron micrographs of isolated human platelets incubated with purified influenza. As one can appreciate the time course, platelets interact with influenza and engulf many influenza viral particles in what has been described as endosomal like structures (See Fig. 2C; Figure 3A). See pg. 7.

Second, we have incubated platelets with influenza, fixed and permeabilized them and stained for intracellular TLR7, influenza and the lysosomal marker CD63 (or LAMP1). We now show in Figure 4E-F and 2B, that certain platelets co-stain with TLR7 and flu, in both in vitro incubated platelets and in vivo in platelets from influenza-infected patients. Co-localization of flu nuclear protein and TLR7 can be seen in Figure 4E-4F,

although it is not evidence of TLR7-activation. As the reviewer is probably aware, TLR7 is activated by viral RNA (not the protein components of the virion) that needs to be released from the viral envelope and the receptor must be at acidic pH in order for activation to occur (as we previously published in Blood, 2014). Additionally, just because one platelet digests the virus it does not mean that it cannot transfer the information to another platelet. Platelets are fluid and contain an open canalicular system throughout which substances can freely be transported. Additionally, platelets constantly communicate with each other (4, 5). As it can be seen in Supplemental Figure 6, platelets can change their morphology as a function of TLR7 stimulation to form long pseudopodia that reach other platelets (Supplemental Figure 6A) and in certain cases the membrane between the pseudopodia and another platelet disappears, suggesting possibility of transfer of information between two platelets (Supplemental Fig 6). We have included this information on pg. 9, 10,

Finally, we want to stress that we do not provide data that the platelet at any point becomes infected with influenza. Since platelets have no nucleus in which viral replication can occur, we can only suggest that platelets sequester influenza from the circulation and digest the virus, thereby leading to activation of the innate immune system. The fact that platelets are not infected is further supported by many studies utilizing human blood which disprove the general occurrence of viremia. We have included all of this information on pg. 16,17.

2) Line 136: we are not convinced by the qRT-PCR that was observed in 50% of patients in 2016. Indeed, qRT-PCR could easily amplify the influenza virus genes from leukocytes contaminate, existing in the samples.

The contamination of leukocytes in our preps is <1 in 50,000. Additionally, we were only able to detect influenza by qPCR when platelets were isolated or were present in the whole blood RNA isolation (Supplemental Table I). There can be a few variables that we cannot control for throughout the years (viral strain, time post infection, severity of infection, etc). We have modified Supplemental Table I to include the samples that tested positive by antibody staining.

3) In figure 1C and D, the labeling of neutrophils is required. Indeed, NETs are known to be released by many cell types such as monocytes (Granger et al. 2017).

We agree that many other cells can release their DNA with *in vitro* treatment of PMA. It is yet to be established if this process can be observed by physiological agonists. Since we have not currently entered flu season, we could not obtain blood from flu patients to stain for neutrophil markers. In order to address the reviewer's question, we treated blood from healthy human donors with flu and stained with CD66. We could only find CD66b positive cells with released DNA (See Figure 1H). Additionally, there are some neutrophils (from an influenza-infected patient) that are releasing their DNA and have the lobular structure, but do not express a detectable CD66b (See Supplemental Fig. 1A). This information is now included on pg. 6.

4) *Figure 1E is not convincing. We agree that the second panel showed the presence of NETs in the blood of infected patients and colocalization with platelets. However the two other panels do not demonstrate these interactions. In the third panel, platelets are not present at all (no green colabeling) and in the fourth panel, the round circle cannot be representative of NETs and the red histones do not colocalize with platelets and DNA.*

We understand how this can be confusing. The point of the 3rd panel was to show platelets in the opposite site of the released DNA. We have included better representation in this version of the manuscript and we are now providing evidence that the released DNA in influenza-infected patients co-localize with CD66b (neutrophil marker, Fig 1G) and MPO (Fig 1E).

5) *Maybe a higher resolution is required for Figure 2B. As shown here, we cannot really distinguish whether the authors point the presence of influenza virus into platelets or simply some vesicles. Accordingly, it is necessary to add a positive control: platelets isolated from a donor and infected in vitro.*

Figure 2B is now 2E. We agree with the reviewer that this figure is not convincing. The higher resolution of Figure 2E makes the morphology of these platelets fuzzy and uninformative. In order for us to solve this problem and to convincingly prove that influenza gets inside of platelets, we are now providing transmission electron microscopy of in vitro influenza-incubated platelets (Fig. 2C, 3A). Additionally, we are including fluorescent images of influenza incubated platelets (Fig 2B, 4F, Supplemental Fig. 5A) that show the virus inside of platelets. The fluorescent microscopy has its own limitations but provides evidence of viral presence in platelets. We have reworded the text to account for this limitation when it comes to Figure 2E. See pg. 7.

6) *Figure 3A: it is necessary to show a figure with all the controls: (i) uninfected platelets that are devoid of TLR7, b) infected platelets that are devoid of TLR7,(iii) uninfected platelets that express TLR7 and (iv) infected platelets that express TLR7.*

Figure 3A is now 4B. We have included new panels that address this comment (also please see the response to comment 8) and provide C3 release as a function of flu and TLR7 stimulation (this is now figure 4B, 4C, Supplemental Figure 5B). In people that have TLR7, flu and Loxoribine (TLR7-agonist) lead to compatible C3 release from the platelets of the same individual. We were only able to find one human donor that consistently lacked TLR7 expression. We repeated the experiment using 3 separate blood draws from this donor, and each time TLR7 was absent and C3 was not released from platelets. See pg. 9.

7) *Figure 6 A: Results for TL7-KO mice need to be shown at 4h and at 24 hours, WT + loxo and TLR7-KO + saline need to be shown.*

Figure 6A is now 8A. We have included representatives of the requested conditions in Figure 8A and Supplemental Figure 11.

8) *it is absolutely required to perform an experiment with infected platelets in presence of an inhibitor of TLR7 and observe an abrogation of C3 release and NETs formation.*

We have performed this experiment and we are including the results in Figure 4. Interestingly, flu or loxoribine induce similar C3 release from platelets and the antagonist (IRS661) is able to inhibit the release of C3 suggesting TLR7-specificity. We have also tested the release of neutrophil-DNA in the presence of platelets and flu, and similarly (Figure 6D) TLR7 antagonism is able to reduce the release of DNA from neutrophils, but was not able to abolish it completely. The latter suggests that other mechanisms may also contribute to the platelet response to influenza. We have included this conclusion on pg. 12.

9) *Figure 6C: why the authors did not performed an anti-neutrophil-Ly6G instead of an anti-H3 (to be homogene with figure 6A)?*

We have and we are including those pictures in Supplementary Fig. 12. The reason is more stylistic as the panel of wt+flu shows beautiful distribution of H4 along the DNA.

In addition, here are minor comments:

- please note in Figure 1, how DNA was labelled

DNA was labeled with DAPI. We have included that information in the legend of Figure 1.

- In line 220, the authors noted no “significant” changes, while in Figure 5A, some differences can be observed.

We agree that, with some people's neutrophils it may seem as they are releasing MPO as a function of TLR7-stimulation. However, although we can isolate pure neutrophils we cannot control for neutrophils that have already formed heterotypic aggregates with platelets or that have engulfed platelets. We also know that human neutrophils do not express TLR7 but TLR8. We have to conclude that even a few platelets can affect the neutrophils in certain isolations. Additionally, companies that provide neutrophil isolation kits by negative depletion do not include CD41 antibody in their depletion kits. We have included this information on pg. 13 and in the methods under “Isolation of human blood components” on pg. 26.

- please add in the material and methods the clones used for anti-CD41 APC and anti-H3 antibodies

We have included the available clones for each antibody used in this study in section “*Confocal microscopy and antibodies*”. See pg. 29.

- *Figure 7B: we do not see the importance of comparing the significance between igG + flu and aPlt + sal. Just add “non significant” between the depleted mice uninfected and infected.*

We have corrected the figure to account for the reviewer's suggestion.

- *line 267: modify the sentence. The virus activates TLR and release of C3, not the platelet*

We have modified the sentence, and the paragraph now reads:

“In this study, we show that influenza is engulfed and recognized by platelet-TLR7, and activation of platelet-TLR7 leads to complement C3 release. Platelet-C3, in turn, pushes neutrophils to release their DNA which is a highly prothrombotic process and may contribute to elevated risk of MI. In support, it is known that, independently of each other, elevated levels of complement C3 and coronary NET burden are predictors of acute MI and myocardial infarct size, respectively (6-8). Our study provides an insight into the mechanism that connects these independent contributors to MI.” See pg. 15.

- *The authors over interpreted the results. While the concept is very interesting, the authors did not show any experiment regarding the elevated risk of myocardial infraction. This should only be briefly discussed.*

We agree with the reviewer. Although we have not shown it ourselves, previous studies show that elevated C3 predicts AMI. We have discussed this on pg. 15 and also please see the response to the previous comment.

Reviewer #3 (Remarks to the Author):

Koupenova et al. conducted a series of experiments using human blood samples and mouse models to identify the relevance of NET formation in the context of platelets, C3, influenza viral infection and myocardial infarction. The authors show that flu patients have DNA-platelet aggregates in the circulation, and platelet TLR7 activation is important for DNA release from neutrophils in vitro. The mouse model data suggest that TLR7 is needed for viral clearance and perhaps important for extracellular trap release.

Following points need to be addressed to strengthen the conclusions made in the manuscript.

Major

1. The intro cites references to highlights the importance of monocytes in MI, but the study focusses on neutrophils.

We agree that this sentence does not focus on the role of neutrophils in MI and we have removed it. To highlight the importance of activated neutrophils and platelet-neutrophil aggregates we have included the following sentence on pg. 4:

“Consistently, neutrophils from the lesion site involved in the initial acute MI are highly activated, form platelet-neutrophil aggregates and can lead to NET burden that is a predictor of ST-segment resolution and the size of MI (7).”

2. No data is provided to support the hypothesis and the relevance of flu, NET formation and myocardial infarction (MI). Therefore, the hypothesis requires a through validation/revision.

We agree with the reviewer that we have not provided direct evidence to support the last postulation of our hypothesis. Previously published studies, however, have provided evidence that:

1. Influenza leads to AMI in the first 7 days post symptom development (9).
2. Elevated levels of plasma complement C3 predict AMI (6, 8)
3. Coronary NET burden are predictors of acute myocardial infarction and myocardial infarct size (7)

Our study provides a novel mechanism that connects these independent contributors to MI during influenza infection. Indeed, future studies are necessary to assess the overall elevation of C3 close to the primary lesion site of MI. To account for the reviewer's concern, we have reworded the connection between our findings and MI. We have discussed these points on pg 15, 18.

3. The paper talks about initial flu infection. Sufficient data are not provided to support the role of initial stage flu infection to DNA release and MI.

The reviewer is correct; we do not have direct data to show that DNA release during flu causes MI. And as stated in the discussion on pg. 15, we cannot assess MI in mice. However, as mentioned in the previous comment, elevated levels of C3 as well as of NETs are independent predictors of acute MI. Since our study shows both, we provide an indirect mechanism that may contribute to MI particularly when the liver becomes overwhelmed and starts to release more C3 as a result of the spiking inflammation during infection. We have changed the language throughout the manuscript.

4. Several technical issues need to be addressed – no clear proof for NETs, image analyses, quantifying NETs and statistical analyses.

We understand the reviewer's concern. In this version, we provide evidence that the DNA released in the influenza-infected patients stems from neutrophils. (Fig. 1F, 1G). Since blood from patients is not synchronized and in certain individuals the damage is severe, we quantified NETosis by treating blood from healthy donors with influenza for 30 min at a dose of 1 virion per 100 platelets. The results are quantified in Figure 1I and the picture representatives are included in 1H. We would like to point out

that our treatments were not administered on plated neutrophils but rather directly in blood at constant rotation of 1000 rpm, which can explain why, in certain cases, the released DNA does not take the form visualized on poly-L-lysine coated plates. We are only testing what would happen directly in the circulation. We have also included quantitation wherever possible such as in Figures 2D, 6E, and 9B.

5. Often an image of a magnified cell is shown to support the data, without quantifying the data set. This is insufficient to make a claim with scientific rigor.

We assume that the reviewer refers to Fig. 2. It is impossible to quantify how many viruses get inside a platelet by confocal fluorescent microscopy (100X, lens), particularly because in certain cases the virus forms aggregates of its own (Figure 3B) or as seen in the TEMs (now included in Fig. 2C and 3A) there is more than one virus per platelet in each endosome-like structure. We are now providing charts that quantify the internalization of the virus by platelets (Fig. 2D) as a function of time using transmission electron microscopy.

6. Several statistical analyses are inappropriate. The t-test cannot be used for comparing more than one set of values. Also, it is incorrect to compare a constant value (control value=1) with other values by t-test. Furthermore, the values that are not significantly different are stated as different (Fig. 7C). Therefore, the conclusions made in several experiments are incorrect.

The manuscript no longer contains graphs that are analyzed by t-test and contain control value of 1. However, in view of the multiple comparisons of different conditions among different people we had to standardize in order to eliminate the variations among donors; these results we analyzed by ANOVA followed by Bonferoni post-test. Figure 7C now Figure 9D, is meant to show only a pattern and not statistically significant increase in the mice that had platelets before infection; we have clearly stated in the text that the difference is not statistically significant (pg.15). We have also removed the p-value from the graph and indicated that is not significant.

7. The claim that the “DNA seen in the circulation is NET” is incorrect because no data is provided to support that the DNA shown in the images belongs to neutrophils (H4 is not a neutrophil marker). The authors also state that the DNA detected in the blood does not contain NET markers. Also, MPO is only “sometimes” present on DNA, suggesting that the process studied here is not NETosis.

We are now providing Figure 1F-1H that provides evidence that the observed release of DNA comes from neutrophils at the beginning of the infection. Neutrophils plated on poly-L-Lys slides are static and traditionally described markers are easier to observe. In our study, in blood (and at constant rotation of 1000 rpm), DNA release is much more dynamic and, depending on time, one can observe traditional markers of NETosis, but certainly these markers can be missed if not caught during the right time frame. Until this study, there was no direct evidence that neutrophils can release their

DNA without attachment. Here we show that platelets are sufficient to induce neutrophil-DNA release but can also control the amount of release. We hope the reviewer is now convinced that a form of NETosis can occur directly in blood.

8. The authors suggest that they have identified a special form of NETosis, but without providing convincing data sets to support their claim.

We apologize if we were not clear about the portrayal of NETosis in the circulation. We do not claim that NETosis in blood is of any special form, rather we provide evidence that neutrophils can release their DNA by signals coming from platelets and that when NETs are analyzed in blood, traditional markers may not be as relevant. Our study makes the platelet-neutrophil relationship an active contributor to viral clearance as well as to the overall prothrombotic burden. We have clarified this on pg. 12

9. C3 is also present in neutrophils. The authors only considered C3 of platelets. Therefore, the interpretation of this data set is also inappropriate.

Neutrophils are fascinating first responders that actually do not contain any components of the complement system, with the exception of one activator of the complement system, properdin, which is released after neutrophil activation (Hallstrom T et al., *Trends in Microbiology* 2010 18, 258-265). In addition, neutrophil TLRs require priming, suggesting that platelets may be an important contributor to neutrophil response to pathogens. We have discussed this on pg. 18

10. TEM images with no immunostaining and quantitative data are not reliable.

We agree with the reviewer, and since TEM immunostaining of platelets is not achievable in our facilities, we now provide Fig 2C (and its quantification in Fig 2D), in which platelets were incubated with influenza in vivo and then subjected to transmission microscopy. The images show internalization of influenza by platelets as almost all stages of phagocytosis can be observed in them.

Minor

11. The introduction is very long. Many points highlighted here belongs to discussion.

We have shortened the introduction.

References:

1. Blair P, Rex S, Vitseva O, Beaulieu L, Tanriverdi K, Chakrabarti S, et al. Stimulation of Toll-like receptor 2 in human platelets induces a thromboinflammatory response through activation of phosphoinositide 3-kinase. *Circ Res*. 2009 Feb 13;104(3):346-54.
2. Koupenova M, Vitseva O, MacKay CR, Beaulieu LM, Benjamin EJ, Mick E, et al. Platelet-TLR7 mediates host survival and platelet count during viral infection in the absence of platelet-dependent thrombosis. *Blood*. 2014 Jul 31;124(5):791-802.
3. Ortiz-Munoz G, Mallavia B, Bins A, Headley M, Krummel MF, Looney MR. Aspirin-triggered 15-epi-lipoxin A4 regulates neutrophil-platelet aggregation and attenuates acute lung injury in mice. *Blood*. 2014 Oct 23;124(17):2625-34.
4. Clancy L, Beaulieu LM, Tanriverdi K, Freedman JE. The role of RNA uptake in platelet heterogeneity. *Thromb Haemost*. 2017 May 3;117(5):948-61.
5. Risitano A, Beaulieu LM, Vitseva O, Freedman JE. Platelets and platelet-like particles mediate intercellular RNA transfer. *Blood*. 2012 Jun 28;119(26):6288-95.
6. Liu D, Qi X, Li Q, Jia W, Wei L, Huang A, et al. Increased complements and high-sensitivity C-reactive protein predict heart failure in acute myocardial infarction. *Biomed Rep*. 2016 Dec;5(6):761-5.
7. Mangold A, Alias S, Scherz T, Hofbauer T, Jakowitsch J, Panzenbock A, et al. Coronary neutrophil extracellular trap burden and deoxyribonuclease activity in ST-elevation acute coronary syndrome are predictors of ST-segment resolution and infarct size. *Circ Res*. 2015 Mar 27;116(7):1182-92.
8. Muscari A, Bozzoli C, Puddu GM, Sangiorgi Z, Dormi A, Rovinetti C, et al. Association of serum C3 levels with the risk of myocardial infarction. *Am J Med*. 1995 Apr;98(4):357-64.
9. Kwong JC, Schwartz KL, Campitelli MA, Chung H, Crowcroft NS, Karnauchow T, et al. Acute Myocardial Infarction after Laboratory-Confirmed Influenza Infection. *N Engl J Med*. 2018 Jan 25;378(4):345-53.

Reviewers' Comments:

Reviewer #1:

Remarks to the Author:

I thank the authors for their discerning comments and addressing our concerns. I think that the quality of work has been improved in this way.

This manuscript is of high interest, original, very easy to read and easy to understand the hypothesis and conclusion of this work.

Now the discussion is in line with the results while taking into account the limitations of the study.

I regret that the authors are unable to respond positively concerning the association of sex on TLR7 expression and an hypothetical modulation of the defense against influenza mediated by platelet-neutrophil cross communication, but I understand the response.

Congratulations for this very informative study.

Reviewer #4:

Remarks to the Author:

The article by Koupenova et al, is a very interesting study that analyzed the effect of influenza infection on platelet-neutrophil mediated-responses.

The most interesting and original data is that the DNA release occurs under constant agitation suggesting that intravascular NETosis can take place in circulation independently of the endothelium. However, a major concern is that traditional markers of NETosis were not used and therefore it is not clear whether these DNA-platelets-neutrophil aggregates represent NETs or not. In fact, the authors refer to these structures ambiguously as NETs or DNA aggregates. In this context, I agree with reviewer #3 concerns and consider that the authors did not address these concerns appropriately. Therefore, in order to be published in Nat Communications, the authors must characterize these DNA structures in a convincing and reliable manner.

Below are my comments regarding NETs and some other issues of the study.

1. To confirm that morphological changes are not due to an artifact of platelet fixation and label, it would be important to add smears of the patient's samples and/or to show mean platelet volume changes in the Coulter to confirm the presence of large platelets.

2. In Fig. 1 A and B It is not clearly visible that platelets contain few granules.

3. How do you differentiate between large (Fig 1 B) from aggregated platelets (Fig 1 C)? Platelets in Fig. B appear to be spread platelets more than large cells as it is stated in the legend. Maybe DIC images could help to clarify this point.

It is not sure that the cell in Fig. 1 C and D is a neutrophil. Please, modify that in the legend.

4. In Fig. 1 D why platelets are not associated with the big mass of DNA?

5. In Fig. 1E it is surprising that the two neutrophils show different staining for H4. If cells were not permeabilized (it is not indicated in M&M) why there is a positive stain in one of the cells? Similar in Fig. 5 A.

6. CD66b is not a NET but rather an activation marker. Citrullinated H3, MPO or elastase are more specific markers of NETosis. It would be important to perform cell labeling with CD66, DAPI and MPO or elastase or H3c to be sure they are NETs.

It is somewhat weird that neutrophils did not express CD66b (Suppl Fig 1A) as it is a surface antigen expressed by most neutrophils. Please, clarify this point.

7. In the control sample of Fig. 1 H, 50% are mixed neutrophil-platelet aggregates which is a high value for a control sample.

8. Please, in order to help the lecture of the article, in page 6 add that released DNA was measured from blood treated with WSN/33 at constant rotation as it is indicated in the legend. In the same page please, correct that influenza led to a 200% increase instead of 40%.

In the correspondent fig 1H for these mentioned experiments, the last picture does not really show that platelets co-localized with the formed NET. In fact, most of the platelets are isolated or aggregated without making contact with neutrophils or the NET. Please, change for a more representative one or delete it.

Again, to confirm they are NETs, neutrophils should be labeled with any of the suggested NET markers.

9. It would be also important to confirm by ELISA the presence of DNA-elastase complexes in the plasma of infected patients or at least nucleosomes.

10. In fig. 4A, the highest value for C3 levels is around 800 ng/ml and in the Suplem Fig 3 it is ten times more. I guess there is an error in some of them.

At the beginning of the paragraph related to TLR7 and the release of C3, there are no comments regarding that some donors do not express this receptor (data obtained from the same group in 2015) and it appears out of the blue in the text. Please, as this is a very important point and to make easier the lecture of the article, it would be helpful if you introduce this concept before the results obtained.

11. Legend of suppl fig 7 does not match with the figure.

In order to improve the reading of the article, suppl Fig. 7 should be combined with Fig. 5. Was GM-CSF measured with cells in suspension and under constant rotation?

It seems to be a problem between the title of Fig 5's legend and the text or the Figure itself since influenza virus does not appear either in the text or the figure. Moreover, according to the text there were just isolated neutrophils so it is not clear the role of platelets in the title and in the figure.

Again, it is important to determine the molecular process that triggered DNA release therefore the experiments should be repeated including NETs markers.

12. It is not clear which experiments were performed under constant agitation and which are not as well as the method used in each experiment to get rid of platelets. Please, clarify this information.

13. Figure 7. Pictures, particularly 7B do not really show DNA with attached platelets, H4 is not a NET marker, and MPO is not observed in the DNA fiber.

Why there are platelet-neutrophil aggregates with the addition of IRSCC1?

14. The fact that neutrophils do not express TLR7 should be further discussed and appropriated references added. Moreover, this concept should be introduced in the correspondent results section.

We thank the reviewers for their expert evaluation of our manuscript. We have addressed all comments, concerns and requests for additional data and believe this has strengthened our study.

Reviewer #1 (Remarks to the Author):

I thank the authors for their discerning comments and addressing our concerns. I think that the quality of work has been improved in this way.

This manuscript is of high interest, original, very easy to read and easy to understand the hypothesis and conclusion of this work.

Now the discussion is in line with the results while taking into account the limitations of the study.

I regret that the authors are unable to respond positively concerning the association of sex on TLR7 expression and hypothetical modulation of the defense against influenza mediated by platelet-neutrophil cross communication, but I understand the response.

Congratulations for this very informative study.

Thank you for these positive comments.

Reviewer #4 (Remarks to the Author):

The article by Koupenova et al, is a very interesting study that analyzed the effect of influenza infection on platelet-neutrophil mediated-responses.

The most interesting and original data is that the DNA release occurs under constant agitation suggesting that intravascular NETosis can take place in circulation independently of the endothelium. However, a major concern is that traditional markers of NETosis were not used and therefore it is not clear whether these DNA-platelets-neutrophil aggregates represent NETs or not. In fact, the authors refer to these structures ambiguously as NETs or DNA aggregates. In this context, I agree with reviewer #3 concerns and consider that the authors did not address these concerns appropriately. Therefore, in order to be published in Nat Communications, the authors must characterize these DNA structures in a convincing and reliable manner.

Below are my comments regarding NETs and some other issues of the study.

We thank the Reviewer for their insightful comments and apologize for the misconception that we think that the neutrophil-DNA released in the circulation is purely due to NETosis. The aim of our study was to describe the process of generation and control of size of platelet-neutrophil-DNA interactions directly in the circulation. The uncontrolled formation of these aggregates ultimately becomes problematic for influenza patients in the first 7 days post infection and increases the risk for myocardial infarction. We aimed to achieve an understanding of the mechanism of platelet contribution to the process of neutrophil-DNA release regardless of the

process of neutrophil-cell death. As the reviewer is aware, NETosis during influenza infection differs from suicidal-NETosis (PMA-mediated) as it is PAD4-independent and negative for citrullinated histone H3. These observations suggest that there are major differences that we cannot account for or solve within the scope of our study. We can say with certainty that there is uniform DNA released from neutrophils that is controlled by platelets during the initial stages of infection. In this sense, throughout our study we have uniformly referred to the released-DNA as “released neutrophil-DNA”.

We have included some of this response in the discussion on p. 21.

1. To confirm that morphological changes are not due to an artifact of platelet fixation and label, it would be important to add smears of the patient's samples and/or to show mean platelet volume changes in the Coulter to confirm the presence of large platelets.

We appreciate the reviewer's comment. We are not able to generate blood smears or determine the MPV for these patients as blood was collected in the past and samples are no longer available for analysis. We also clarify that the word “large” did not mean a large platelet but was meant to be read with the full statement of “large flattened satellite-like platelets”. As the reviewer can surmise, these are not actually large platelets in the traditional sense. We have removed “large” from the sentence to prevent further confusion. The sentence reads now (see p. 6):

“...ranging from small platelets without pseudopodia, to spread-out, flattened, satellite-like platelets that in some cases have a diameter bigger than 10 μm (Figure 1A, 1B, Supplemental Figure 1A).

To address the reviewer's concern about fixation and labeling, we controlled for the effects of fixation using uninfected controls and processed the blood from both groups identically. Antibody labeling was always done after fixation to avoid unwanted effects of the staining on platelets. The observed platelet-morphology was only present in the influenza-infected patients. Additionally, platelet spreading and flattening is a known phenomenon during adhesion (PMID: 511936). We have included this information in the methods on p. 29.

2. In Fig. 1 A and B It is not clearly visible that platelets contain few granules.

We apologize for this misunderstanding. The reviewer is completely correct as we have not done any granule staining to make the claim. We have changed the sentence to: “...ranging from small platelets without pseudopodia, to flattened satellite-like platelets that in some cases have a diameter bigger than 10 μm (Figure 1A, 1B, Supplemental Figure 1A)” (see p. 6).

3. How do you differentiate between large (Fig 1 B) from aggregated platelets (Fig 1 C)?

Platelets in Fig. B appear to be spread platelets more than large cells as it is stated in the legend. Maybe DIC images could help to clarify this point.

We thank the reviewer for catching the problem with this statement. The reviewer is completely correct; the platelets in Figure 1C cannot be distinguished from 1B because they are the same large flattened platelets. The aggregates that we were referring to were platelet-DNA aggregates, not solely platelet aggregates. To avoid this misconception, we have corrected the sentence to: "Surprisingly, the blood of influenza-infected patients contained aggregates of released DNA associated with (flattened) platelets (Figure 1C-1E)" (see p. 6). We are now also including Supplemental Figure 1A that includes the DIC which is quite different than the DIC of platelet-aggregates (See Supplemental Figure 4B). Of note, DIC of spread-out platelets cannot be captured well by confocal microscopy.

It is not sure that the cell in Fig. 1 C and D is a neutrophil. Please, modify that in the legend.

We have made that correction in the figure legend on p. 48.

4. In Fig. 1 D why platelets are not associated with the big mass of DNA?

This is a great question to which we can only speculate that perhaps, in addition to reduced platelet count observed during influenza infection (discussed on p. 18), after the initial burst of interaction in the influenza-infected patients' platelets, the platelets may become dysfunctional and unable to recognize the exposed DNA similarly to the initial stages as in Figure 1H. We have included the following sentence in the discussion on p. 18:

"Furthermore, the satellite platelet morphology may also represent a dysfunctional platelet population that is unable to interact properly with released DNA as evidenced by the spread-out interaction between platelets and the DNA in our fluorescent images."

5. In Fig. 1E it is surprising that the two neutrophils show different staining for H4. If cells were not permeabilized (it is not indicated in M&M) why there is a positive stain in one of the cells? Similar in Fig. 5 A.

The reviewer is correct that H4 staining is visualized in non-permeabilized neutrophils. A problem with using human blood as a control is that these individuals do not live in a sterile environment and are under constant exposure to potential pathogens that can be undetected. We have included these pictures of control neutrophils from healthy donors to point out that they are present in the circulation of human donors.

6. CD66b is not a NET but rather an activation marker. Citrullinated H3, MPO or elastase are more specific markers of NETosis. It would be important to perform cell labeling with CD66, DAPI and MPO or elastase or H3c to be sure they are NETs.

The reviewer makes an important point and we have done an additional experiment to address the reviewer's concern. We are now including images in Figure 1K (last row), in addition to 1F and 1G (1F and 1G are images of the same aggregate), showing that the DNA, CD66b and MPO coincide.

It is somewhat weird that neutrophils did not express CD66b (Suppl Fig 1A) as it is a surface antigen expressed by most neutrophils. Please, clarify this point.

This is a valid observation but clarification as to why certain neutrophils may not express CD66b is experimentally beyond the scope of the current study. There is a possibility that certain neutrophils release CD66b and this cell adhesion molecule can no longer be detected on the surface. In support, evidence for secretion of CD66b as a function of infection has been previously observed by other groups (PMID: 24743304; 15541289). To clarify this observation, the following sentence is now in the main text on p. 6:

“In the influenza-infected patients, certain DNA-releasing neutrophils did not express CD66b (Supplemental Figure 1B), indicating that some of the released DNA cannot be absolutely identified as coming from neutrophils. Lack of CD66b may be due to secretion of this adhesion molecule during infection as previously observed (14, 18).”

7. In the control sample of Fig. 1 H, 50% are mixed neutrophil-platelet aggregates which is a high value for a control sample.

These are *in vitro* experiments that no longer have the beneficial platelet-function regulating effect on platelets, and interactions are observed as a function of time. In this study we have treated blood only for a short period of time (no longer than 40 min and immediately after blood draw). Of note, baseline heterotypic aggregates are present at various levels in uninfected donors. Additionally, as mentioned in comment 5, we are unable to control for pathogen exposure of our non-symptomatic human donors. In order for us to solve this variability, analysis in all cases was done as a function of control blood treated with the buffer in which the virus and agonists are dissolved.

8. Please, in order to help the lecture of the article, in page 6 add that released DNA was measured from blood treated with WSN/33 at constant rotation as it is indicated in the legend.

We have included the following sentence on p. 6: “Of note, the released DNA was measured in blood from healthy donors treated with WSN/33 at constant rotation.”

In the same page please, correct that influenza led to a 200% increase instead of 40%.

We appreciate the reviewer’s suggestion. However our quantitative data shows that the increase is ~40% not 200%, as we see only 40% of the entire neutrophil population to be releasing DNA.

In the correspondent fig 1H for these mentioned experiments, the last picture does not really show that platelets co-localized with the formed NET. In fact, most of the platelets are isolated or aggregated without making contact with neutrophils or the NET. Please, change for a more representative one or delete it.

Our aim is to show representatives of various interactions between the neutrophil-DNA and platelets. In the case to which the reviewer is referring, the DNA is not entirely covered with platelets, but platelets are interacting with it, suggesting either particular kinetics of interaction and/or a physiological relationship that needs further characterization beyond the scope of our study. We have included a comment in the legend of Figure 1.

Again, to confirm they are NETs, neutrophils should be labeled with any of the suggested NET markers.

We are now including a panel in 1F and 1G that has co-staining of CD66b, MPO and DAPI.

9. It would be also important to confirm by ELISA the presence of DNA-elastase complexes in the plasma of infected patients or at least nucleosomes.

We have confirmed that influenza patients have elevated Elastase and MPO in the plasma but we could not detect changes in nucleosomes or DNA levels. Having in mind that in vivo neutrophil-DNA release is a bit different than PMA-mediated NETosis, and that platelets and DNA form aggregates, most likely our double-spinning method of processing ACD-plasma immediately after blood draw is not the most appropriate for testing NETosis in plasma. Our fluorescent microscopy data in Figure 1 in combination with the newly included ELISA for Elastase and MPO (Figure 1H, 1I) however provide evidence that released neutrophil-DNA is found in the circulation during stimulation with influenza or in the influenza-infected patients.

10. In fig. 4A, the highest value for C3 levels is around 800 ng/ml and in the Supplem Fig 3 it is ten times more. I guess there is an error in some of them.

We thank the reviewer for pointing this out. We have corrected the supplemental figure to the appropriate concentration.

At the beginning of the paragraph related to TLR7 and the release of C3, there are no comments regarding that some donors do not express this receptor (data obtained from the same group in 2015) and it appears out of the blue in the text. Please, as this is a very important point and to make easier the lecture of the article, it would be helpful if you introduce this concept before the results obtained.

That is a good suggestion. We have included the following sentence at the beginning of the section on p. 9:

“Lack of expression of TLR7-mRNA in platelets, however, has been observed in some human donors (11).”

11. Legend of suppl fig 7 does not match with the figure.

We thank the reviewer for pointing this out. We have corrected the legends of the Supplemental figures to correspond to the actual figures.

In order to improve the reading of the article, suppl Fig. 7 should be combined with Fig. 5. Was GM-CSF measured with cells in suspension and under constant rotation?

We have combined the two figures into one. All experiments were done under constant rotation and we have now included that information in the legend of Figure 5 and in the Methods section.

It seems to be a problem between the title of Fig 5's legend and the text or the Figure itself since influenza virus does not appear either in the text or the figure. Moreover, according to the text there were just isolated neutrophils so it is not clear the role of platelets in the title and in the figure.

We have corrected the figure legend to state:

Figure 5: Platelets secrete GM-CSF in the presence of neutrophils and TLR7 stimulation; GM-CSF abrogates the effect of Complement C3 mediated-neutrophil-DNA release.

Again, it is important to determine the molecular process that triggered DNA release therefore the experiments should be repeated including NETs markers.

We agree with the reviewer, when it comes to blood, that we need to definitively show that neutrophils are the cells releasing DNA. The neutrophils in Figure 5 and all other in vitro experiments are isolated neutrophils that have less than 2% eosinophils. In that sense, the DNA release, which is overwhelming, would mostly come from neutrophils.

12. It is not clear which experiments were performed under constant agitation and which are not as well as the method used in each experiment to get rid of platelets. Please, clarify this information.

All experiments are performed under constant rotation (agitation is not appropriate for platelet experiments). We have now included a sub-section in the Methods that states the conditions:

"In vitro experimental conditions

All in vitro experiments in this study were performed at 37°C and constant rotation of 1000 rpm in a PAP8 aggregometer for the indicated time."

When it comes to eliminating platelets from human blood, we have listed a reference to the Ficoll protocol, and in the methods starting on p. 26 we have listed exactly the materials and washes that we used to achieve the separation. We have also included a crude blood count by the two methods in Supplemental Figure 8.

13. Figure 7. Pictures, particularly 7B do not really show DNA with attached platelets, H4 is not a NET marker, and MPO is not observed in the DNA fiber.

Figure 7B shows the release of MPO from neutrophils in the presence of platelets. We were not looking at DNA in this figure.

Why there are platelet-neutrophil aggregates with the addition of IRS661?

This is a great question. Influenza is a single-stranded RNA virus that can also activate in people who express TLR8 in their platelets. Also, inhibitors are not perfect and do not achieve 100% inhibition nor are they 100% specific.

14. The fact that neutrophils do not express TLR7 should be further discussed and appropriated references added. Moreover, this concept should be introduced in the correspondent results section.

We have included the following sentence on p. 8: "Platelets express functional TLR7 and TLR7 stimulation leads to alpha granule protein release (6) while neutrophils do not express TLR7 at the protein level (22)."

Reviewers' Comments:

Reviewer #4:

Remarks to the Author:

In the revised version of the article by Milka Koupenova et. al, the authors have addressed most of my concerns. Thank you for considering my suggestions which have clarified many issues. However, there are still some points which need to be clarified.

1) The Fig 1 E-G is still not very clear for me. You answered that Fig 1E was not permeabilized for staining of H4, so I understand you are considering that H4 positive signal is on the cellular surface and that could be used as an indicator of NETosis including in healthy donors?

I understand that donors do not live in a sterile environment, but most of the neutrophils usually are non-activated in healthy donors and if that happens, that sample should not be included in the study as control of healthy donors. The figure shows two cells and one is activated.

Since in Fig. 1-G all cells are positive for MPO and it is a granular enzyme I guess that cells are permeabilized. Is that correct?

Please, clarify in the legend whether control cells were permeabilized or not.

2) Fig 8. How did you quantify the released-DNA in whole blood? How do you know that the released-DNA is coming exclusively from neutrophils and not from monocytes, lymphocytes, or dead cells?

We again thank the Reviewer for their expert and careful evaluation of our manuscript. We have addressed all comments, concerns and requests for additional data and believe this has strengthened and clarified our study.

Reviewer #4 (Remarks to the Author):

In the revised version of the article by Milka Koupenova et. al, the authors have addressed most of my concerns. Thank you for considering my suggestions which have clarified many issues. However, there are still some points which need to be clarified.

1) The Fig 1 E-G is still not very clear for me. You answered that Fig 1E was not permeabilized for staining of H4, so I understand you are considering that H4 positive signal is on the cellular surface and that could be used as an indicator of NETosis including in healthy donors?

I understand that donors do not live in a sterile environment, but most of the neutrophils usually are non-activated in healthy donors and if that happens, that sample should not be included in the study as control of healthy donors. The figure shows two cells and one is activated.

The reviewer makes an important point and we understand how this may not be an optimal representation of the in vivo state of neutrophils in a healthy individual. We are not comfortable claiming that the surface H4 positive signal on a neutrophil with defined lobes is an indicator of NETosis but we cannot exclude some form of underlying activation. In some control-donors H4 is not easily detected in the neutrophils. There is also a possibility that since not all neutrophils are the same, some of them, as they stay in formaldehyde (blood from our patients was stored for a few days before staining), may become more permeable to the antibody. It is known that formaldehyde as it cross-links with the proteins in the cell can lead to “indentations and vacuoles forming on or close to the nuclear and mitochondrial membranes”, that allows for the antibody to have easy access to the intracellular content of the cell (1, 2). Because we think that the H4 positive signal in the image is more of an experimental (rather than physiological) artifact and we do not see this uniformly among donors, we have taken the reviewer’s suggestion and included a different representative image. We have included this limitation in the legend of Figure 1.

Since in Fig. 1-G all cells are positive for MPO and it is a granular enzyme I guess that cells are permeabilized. Is that correct? Please, clarify in the legend whether control cells were permeabilized or not.

We stained for MPO after prolonged storage of the collected blood in formaldehyde and although we did not permeabilize, clearly some of the control neutrophils stain more positively than others. This is not the case when we fix for only 10 min as seen in the control blood included in Figure 1K. We have included another image from the blood of a different donor that

did not stain as bright but still had some positive staining for MPO. We have included a sentence in the legend of Figure 1 stating:

“Blood from influenza-infected patients and controls in (Figures) 1A-1G was not permeabilized. Since formaldehyde can cause certain levels of permeabilization as a function of cross-linking, positive staining in control samples for H4 and MPO do not necessarily indicate activation.”

2) *Fig 8. How did you quantify the released-DNA in whole blood? How do you know that the released-DNA is coming exclusively from neutrophils and not from monocytes, lymphocytes, or dead cells?*

The reviewer makes an important point. In Figure 8A and B, the quantitation was based only on aggregates that stained for both Ly6G and DAPI. Ly6G stains brightly only for neutrophils and eosinophils; monocytes and lymphocytes did not exhibit positive staining. In order to eliminate any confusion, we have changed the text and re-labeled the graphs to state “DNA release from Ly6G-positive cells”. We have done this in Figure 9, Supplemental Figure 10 and 11 as well.

References:

1. Fox CH, Johnson FB, Whiting J, Roller PP. Formaldehyde fixation. *J Histochem Cytochem.* 1985 Aug;33(8):845-53.
2. Hobro AJ, Smith NI. An evaluation of fixation methods: Spatial and compositional cellular changes observed by Raman imaging. *Vibrational Spectroscopy.* 2017 2017/07/01;91:31-45.